# Evaluating the E3SM Land Model version 0 (ELMv0) at a temperate forest site using flux and soil water measurements

Junyi Liang[1,*], Gangsheng Wang[1,2,*], Daniel M. Ricciuto[1], Lianhong Gu[1], Paul J. Hanson[1], Jeffrey D. Wood[3], Melanie A. Mayes[1]

[1]Environmental Sciences Division & Climate Change Science Institute, Oak Ridge National Laboratory, Oak Ridge, TN 37831, USA
[2]Department of Microbiology and Plant Biology, University of Oklahoma, Norman, Oklahoma, 73019, USA
[3]School of Natural Resources, University of Missouri, Columbia, MO, 65211, USA

*Correspondence to*: Junyi Liang (email: liangj@ornl.gov) and Gangsheng Wang (email: wanggs@ou.edu)

**Abstract.** Accurate simulations of soil respiration and carbon dioxide ($CO_2$) fluxes are critical to project global biogeochemical cycles and the magnitude of carbon-climate feedbacks in Earth system models (ESMs). Currently, soil respiration is not represented well in ESMs, and few studies have attempted to address this deficiency. In this study, we evaluated the simulation of soil respiration in the Energy Exascale Earth System Model (E3SM) Land Model version 0 (ELMv0) using long-term observations from the Missouri Ozark AmeriFlux (MOFLUX) forest site in the central U.S. Simulations using the default model parameters underestimated soil water potential (SWP) during peak growing seasons and overestimated SWP during non-growing seasons, and consequently underestimated annual soil respiration and gross primary production (GPP). A site-specific soil water retention curve greatly improved model simulations of SWP, GPP and soil respiration. However, the model continued to underestimate the seasonal and interannual variabilities and the impact of the extreme drought in 2012. Potential reasons may include inadequate representations of vegetation mortality, the soil moisture function, and the dynamics of microbial organisms and soil macroinvertebrates. Our results indicate that the simulations of mean annual GPP and soil respiration can be significantly improved by better model representations of the soil water retention curve.

## 1 Introduction

Globally, soils store over twice as much carbon (C) as the atmosphere (Chapin III et al., 2011). Soil respiration (SR) is the second largest C flux between terrestrial ecosystems and the atmosphere (Luo and Zhou, 2006). An accurate simulation of SR is critical for projecting terrestrial C status, and therefore climate change, in Earth system models (ESMs) (IPCC, 2013).

Despite significant experimental data accumulation and model development during the past decades, simulations of soil $CO_2$ efflux to the atmosphere still have a high degree of uncertainty (Friedlingstein et al., 2006; Jones et al., 2013; Todd-Brown et al., 2013; Todd-Brown et al., 2014; Tian et al., 2015), calling for comprehensive assessments of model performance against observational data.

To assess the performance of ESMs, different types of data can be used. For example, using atmospheric $CO_2$ observations, eddy covariance measurements and remote sensing images, Randerson et al. (2009) found that two ESMs underestimated net C uptake during the growing season in temperate and boreal forest ecosystems, primarily due to the delays in the timing of maximum leaf area in the models. By comparing remote sensing estimations from the Moderate Resolution Imaging Spectroradiometer and flux tower datasets, Xia et al. (2017) found that better representations of processes controlling monthly maximum gross primary productivity (GPP) and vegetation C use efficiency (CUE) improved the ability of models to predict the C cycle in permafrost regions.

Despite the significance of large global SR fluxes, SR has rarely been evaluated in ESMs using long-term observations. Among the factors that influence SR, soil water potential (SWP) provides a unified measure of the energy state of soil water that limits the growth and respiration of plants and microbes. Unlike soil temperature (ST) or soil volumetric water content (VWC), however, SWP is difficult to directly monitor in the field. Accurate estimation of SWP largely relies on the soil water retention curve (i.e., the relationship between VWC and SWP), which is highly specific to soil properties (Childs, 1940; Clapp and Hornberger, 1978; Cosby et al., 1984; Tuller and Or, 2004; Moyano et al., 2013). Site-level data have been used to evaluate model representations of other processes, such as phenology, net primary production (NPP), transpiration, leaf area index (LAI), water use efficiency, and nitrogen use efficiency (Richardson et al., 2012; De Kauwe et al., 2013; Walker et al., 2014; Zaehle et al., 2014; Mao et al., 2016; Duarte et al., 2017; Montané et al., 2017). In Powell et al. (2013), the only aspect influencing the modelling of SR was the sensitivity of SR to VWC in an Amazon forest, but the study resulted in no improvements to simulated SR. Here, we focus on improving simulations by using site-specific measurements to assess multiple factors influencing SR.

We will evaluate the simulation of SR step by step. We assessed underlying mechanisms in the Energy Exascale Earth System Model (E3SM) Land Model version 0 (ELMv0) by using intensive observations at the Missouri Ozark AmeriFlux (MOFLUX) forest site in the central U.S. We first evaluated the effects of two abiotic factors, ST and SWP, on the simulation SR. Then we evaluated the effects of biotic factors, such as GPP, LAI and $Q_{10}$ of heterotrophic respiration, on the simulation of surface $CO_2$ efflux to the atmosphere.

## 2 Materials and Methods

### 2.1 Study site and measurements

The MOFLUX site is located in the University of Missouri's Thomas H. Baskett Wildlife Research and Education Area (latitude 38º44'39"N, longitude 92º12'W). The mean annual precipitation is 1083 mm, while minimum and maximum monthly mean temperatures are −1.3 ºC (January) and 25.2 ºC (July), respectively. The site is a temperate, upland oak-

hickory forest, with major tree species consisting of white oak (*Quercus alba* L.), black oak (*Q. velutina* Lam.), shagbark hickory (*Carya ovata* (Mill.) K. Koch), sugar maple (*Acer saccharum* Marsh.), and eastern red cedar (*Juniperus virginiana* L.) (Gu et al., 2016; Wood et al., 2017). The dominant soils are the Weller silt loam and the Clinkenbeard very flaggy clay loam (Young et al., 2001).

5      Ecosystem C, water and energy fluxes, SR, LAI and supporting meteorolgecal measurements were initiated in June 2004 (Gu et al., 2016). Soil respiration was measured within the ecosystem flux tower footprint using non-flow through non-steady state auto-chambers. From 2004 through 2013, SR was measured using eight automated, custom-built chambers (ED system; Edwards and Riggs, 2003; Gu et al., 2008) coupled with an infrared gas analyzer (LI-820 Li-Cor Inc., Lincoln, Nebraska). In 2013, this system was replaced with 16 auto-chambers operated using the closed-path system (model LI-8100; 10  Li-Cor Inc., Lincoln, Nebraska). The two systems (ED and Li-8100) were operated side-by-side for several weeks in 2010 and found to produce comparable responses (Paul Hanson, personal communication).  Half-hourly SR time series were generated to coincide with the ecosystem flux data set by averaging those chambers sampled in the corresponding averaging period. Net ecosystem $CO_2$ exchange (NEE) was measured on a 32-m walk-up scaffold tower (Gu et al., 2016). A soil temperature profile sensor (model STP01, HuksefluxUSA, Inc., Center Moriches, NY) measured at 5 depths down to 0.5 m. 15  Soil VWC was measured using water content reflectometers (model CS616, Campbell Scientific Inc., Logan UT) installed beneath each soil chamber. All the data were recorded at half-hourly intervals, which were integrated over time to obtain daily and annual fluxes.

**2.2 Ecosystem C flux partitioning**

Flux-tower GPP was estimated from measured NEE. To reduce biases resulting from individual methods, three NEE-20  partitioning approaches were employed. The average and variation of the three methods were used to evaluate the model-simulated GPP. In the first two methods, ecosystem respiration (ER) was estimated from nighttime NEE and extrapolated to daytime, and daytime GPP was calculated from NEE and the extrapolated ER (Reichstein et al., 2005). The only difference between the two methods was whether to exclude night-time data under non-turbulent conditions. In the third method, GPP was estimated by fitting the light-response curve between NEE and radiation (Lasslop et al., 2010). All the partitioning 25  calculations were conducted using the R package *REddyProc* (Reichstein et al., 2017).

**2.3 Model description**

The ELMv0 used in this study is structurally equivalent to the Community Land Model 4.5 (CLM 4.5), which includes coupled carbon and nitrogen cycles (Oleson et al., 2013). In ELMv0, the soil biogeochemistry can be simulated with one-layer or multi-layer converging trophic cascade (CTC, i.e., CLM-CN) decomposition model. We used the vertically-resolved 30  CTC decomposition in this study. In the model, SR was calculated by different $CO_2$ emission components (Oleson et al., 2013):

$$SR = R_A + R_H \qquad \text{Eq. (1)}$$
$$R_A = R_M + R_G \qquad \text{Eq. (2)}$$

$$R_M = R_{livecroot} + R_{froot} \qquad \text{Eq. (3)}$$

$$R_{liveCroot} = [N]_{liveCroot} R_{base} R_{q10}^{(T_{2m}-20)/10} \qquad \text{Eq. (4)}$$

$$R_{froot} = \sum_{j=1}^{10} [N]_{froot} rootfr_j R_{base} R_{q10}^{(T_{2m}-20)/10} \qquad \text{Eq. (5)}$$

$$R_G = 0.3 C_{new\_root} \qquad \text{Eq. (6)}$$

$$R_H = \sum_{j=1}^{10} \sum_{i=1}^{4} SOC_{ij} k_i rf_i \xi_T \xi_W \xi_O \xi_D \xi_N \qquad \text{Eq. (7)}$$

where $R_A$ and $R_H$ are belowground autotrophic and heterotrophic respiration, respectively. $R_A$ is the sum of root maintenance ($R_M$) and growth respiration ($R_G$). $R_{livecroot}$ and $R_{froot}$ are maintenance respiration of live coarse root and fine root. $[N]_{livecroot}$ and $[N]_{froot}$ are nitrogen content of live coarse and fine roots. $R_{base}$ is the base maintenance respiration at 20 ºC. $R_{q10}$ which equals 2, is the temperature sensitivity of maintenance respiration. $T_{2m}$ is the air temperature at 2 m. $C_{new\_root}$ is the new root growth C. $R_H$ is the sum of heterotrophic respiration of four SOC pools with different turnover rates (Oleson et al., 2013) in the 10 soil layers. The parameters $k_i$ and $rf_i$ are the turnover rate and respiration fraction of the $i^{th}$ pool. $\xi_T$, $\xi_W$, $\xi_O$, $\xi_D$, $\xi_N$ are environmental modifiers of soil temperature, soil water content, oxygen, depth and nitrogen for each layer, respectively. A detailed description of the environmental modification can be found in Oleson et al. (2013). Briefly, the temperature and water modifiers were:

$$\xi_T = Q_{10}^{\left(\frac{T_{soil}-T_{ref}}{10}\right)} \qquad \text{Eq. (8)}$$

$$\xi_W = \begin{cases} 0 & for\ \Psi < \Psi_{min} \\ \dfrac{log(\Psi_{min}/\Psi_m)}{log(\Psi_{min}/\Psi_{max})} & for\ \Psi_{min} \leq \Psi \leq \Psi_{max} \\ 1 & for\ \Psi > \Psi_{max} \end{cases} \qquad \text{Eq. (9)}$$

where $Q_{10}$ is the temperature sensitivity (the default value is 1.5), $T_{ref}$ is the reference temperature (25 ºC). $\Psi_m$ is the matric water potential, $\Psi_{min}$ is the lower limit for matric potential, and $\Psi_{max}$ is the matric water potential under saturated conditions. The ELMv0 is a grid-based model. To assess it using site-level observations, we used a point-run framework which allows the model to simulate individual sites (Mao et al., 2016). Single-point runs forced with site-level measurements have a long history to evaluate model representations of phenology, NPP, transpiration, LAI, water use efficiency, and nitrogen use efficiency (Richardson et al., 2012; De Kauwe et al., 2013; Walker et al., 2014; Zaehle et al., 2014; Mao et al., 2016; Duarte et al., 2017; Montané et al., 2017). With site-specific forcing, a 200-year accelerated decomposition spin-up was performed, followed by a 200-year normal spin-up, before the transient simulation was performed from 1850 to 2013. The vegetation was set as 100% temperate deciduous forest.

## 2.4 Soil water retention curve

Soil water potential values for the Weller soils (https://soilseries.sc.egov.usda.gov/OSD_Docs/W/WELLER.html) were estimated from observed VWC and soil water retention curves that were developed for the site. To derive the soil water retention curves, soil samples were collected in the area of the flux tower base at two depths: 0 to 30 cm and below 30 cm.

5  Samples were evaluated periodically for soil water potential using a dewpoint potentiometer (Decagon Devices, Model WP4C) as they dried over time (Hanson et al., 2003).

In the ELMv0, the SWP was calculated from VWC based on the Clapp & Hornberger model (Clapp and Hornberger, 1978), in which the SWP-VWC relationship was expressed as

$$\Psi_m = \Psi_s \left(\frac{\theta}{\theta_s}\right)^{-B} \qquad \qquad \text{Eq. (10)}$$

10  where $\theta$ and $\Psi_m$ are the VWC and matric potential (MPa); and $\theta_s$ and $\Psi_s$ are VWC and matric potential under saturated conditions, and $B$ is a parameter to determine the shape of the SWP-VWC relationship. In the ELMv0, all parameters were calculated from the fraction of organic matter ($f_{om}$), clay content ($f_{clay}$; %) and sand content ($f_{sand}$; %) (Cosby et al., 1984; Lawrence and Slater, 2008), where

$$\Psi_s = -\left((1 - f_{om}) \times 10 \times 10^{1.88 - 0.0131 f_{sand}} + 10.3 f_{om}\right) \qquad \text{Eq. (11)}$$

$$\theta_s = \left((1 - f_{om}) \times (0.489 - 0.00126 f_{sand}) + 0.9 f_{om}\right) \qquad \text{Eq. (12)}$$

$$B = (1 - f_{om}) \times \left(2.91 + 0.159 f_{clay}\right) + 2.7 f_{om} \qquad \text{Eq. (13)}$$

In addition to the Clapp & Hornberger model, four other empirical models (Brooks and Corey, 1964; van Genuchten, 1980; Fredlund and Xing, 1994; Hanson et al., 2003) were also used to fit the SWP curve against VWC (Table 1, Figure 1).

In the Brooks & Corey model, the SWP-VWC relationship was expressed as

$$\frac{\theta - \theta_r}{\theta_s - \theta_r} = \begin{cases} \left(\frac{\Psi_b}{\Psi_m}\right)^{\lambda} & \Psi_m > \Psi_b \\ 1 & \Psi_m \leq \Psi_b \end{cases} \qquad \text{Eq. (14)}$$

where $\theta_r$ and $\theta_s$ are the residual and saturated water contents, respectively, $\theta$ and $\Psi_m$ are measured VWC and matric potential (MPa), $\Psi_b$ is a parameter related to the soil matric potential at air entry, and $\lambda$ is related to the soil pore size distribution (Brooks and Corey, 1964).

In the Fredlund & Xing model, the SWP-VWC relationship was described as

25  
$$\frac{\theta - \theta_r}{\theta_s - \theta_r} = \left[\frac{1}{\ln\left(e + (\Psi_m/a)^n\right)}\right]^m \qquad \text{Eq. (15)}$$

where $a$, $n$ and $m$ are parameters determining the shape of the soil water characteristic curve (Fredlund and Xing, 1994).

In the Hanson model (Hanson et al., 2003), soil matric potential was modelled by a double exponential function:

$$\Psi_m = -a^{b\theta^c} - d \qquad \qquad \text{Eq. (16)}$$

where $a$, $b$, $c$ and $d$ are fitted parameters.

30  In the van Genuchten model, the SWP-VWC relationship was described as

$$\frac{\theta - \theta_r}{\theta_s - \theta_r} = \left[\frac{1}{1 + (\alpha \Psi_m)^n}\right]^{(1-1/n)} \qquad \text{Eq. (17)}$$

where $\alpha$ (MPa$^{-1}$) and $n$ are parameters that determine the shape of the soil-water curve (van Genuchten, 1980).

In addition to the default SWP-VWC relationship in the ELMv0, all the five empirical models were parameterized using non-linear fitting against measured VWC and SWP data from the study site. For the calibration of the Clapp & Hornberger model, instead of using the hard-coded parameters in Eq. 11-13, we calibrated the three parameters (i.e., $\Psi_m$, $\theta_s$ and $\Psi_s$) in the Clapp & Hornberger model (Eq. 10). The root-mean-square error (RMSE) and Akaike Information Criterion (AIC) were used to select the best model representing the SWP-VWC relationship. The AIC value was calculated by:

$$AIC = a ln\left(\frac{\sum(\hat{\varepsilon})^2}{a}\right) + 2b \qquad \text{Eq. (18)}$$

where a is the number of data points, $\hat{\varepsilon}$ is the estimated residual of each data point, and b is the total number of estimated model parameters. Smaller RMSE and AIC values imply a better fit to observational data. The best-fit model was used in two ways. First, it was used to calculate the "observed" SWP from monitored VWC in the field. Second, it was implemented in the ELMv0 to replace the default SWP model in order to improve the SWP simulation.

## 2.5 Evaluation of SR in the model

The evaluation of SR was conducted step by step. We first compared observations with the model default output of SR and related factors, including ST, SWP, GPP, and LAI. Thereafter, we attempted to improve the simulation of these factors in order to improve the overall SR simulation by (i) implementing the best-fit SWP-VWC relationship, and (ii) modifying model parameters related to GPP, LAI and SR. GPP-related parameters included the specific leaf area (SLA) at the top of canopy and the fraction of leaf nitrogen in the RuBisCO enzyme. LAI-related parameters included the number of days to complete leaf fall during the end of growing season, the critical day length for senescence (i.e., the length of the day when leaves start to senesce), and a parameter $\alpha$ that was used to produce a linearly-increasing rate of litterfall. The contributions and autotrophic and heterotrophic respiration to total SR were also calculated. In addition, the Q$_{10}$ of heterotrophic respiration was also modified. Because the parameter modification was dependent on the evaluation steps, how the parameters were modified is presented in the Results section.

## 3 Results

For the upper 30 cm of soil, the ELMv0 simulations using the default Clapp and Hornberger model tended to underestimate the SWP when VWC was less than 15% (Fig. 1a), while SWP rapidly approached zero when VWC was greater than 25% (Fig. 1a). For soil below 30 cm, the ELMv0 showed a consistent overestimation of SWP (Fig. 1b). The default ELMv0 showed relatively high RMSE for both soil layers, indicating that the SWP-VWC relationship was not well simulated in the ELMv0 (Table 1). Although the Clapp & Hornberger model performed better by using parameters from non-linear fitting, its

performance was not as good as the Hanson and the van Genuchten models (Table 1, Fig. 1). The Hanson model was the best-fit model for the MOFLUX site, showing the smallest RMSE and AIC values for both soil layers (Table 1, Fig. 1), and was therefore implemented in ELMv0 to calculate SWP from measured VWC.

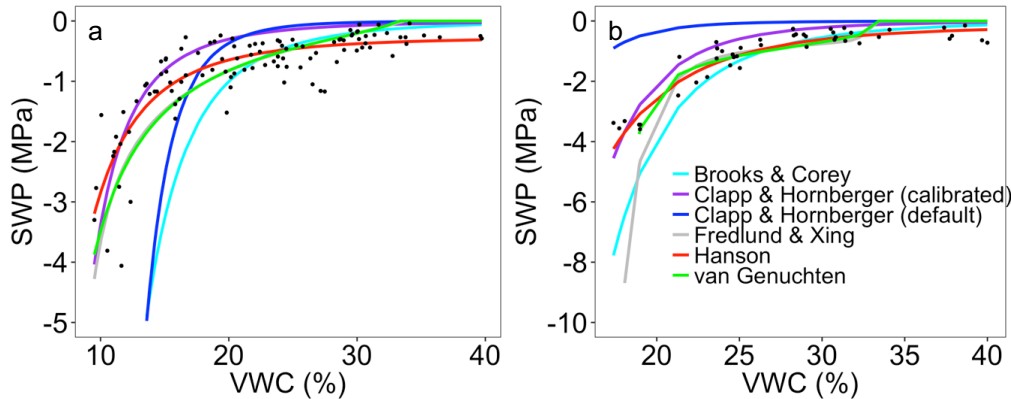

**Figure 1: Observed (black dots) and simulated relationship between soil water potential (SWP) and volumetric water content (VWC) by the different models at two soil layers: (a) 0 to 30 cm and (b) below 30 cm.**

**Table 1.** Root-mean-square-error (RMSE) and Akaike Information Criterion (AIC) of different models in simulating the SWP-VWC relationship for the soil in the MOFLUX site at two depths: 0 to 30 cm and below 30 cm.

| Model | < 30 cm | | > 30 cm | |
|---|---|---|---|---|
| | RMSE | AIC | RMSE | AIC |
| Clapp & Hornberger (default ELMv0) | 4.25 | 157.82 | 1.33 | 18.51 |
| Brooks & Corey | 3.91 | 151.05 | 1.13 | 13.51 |
| Clapp & Hornberger (calibrated) | 0.53 | -61.03 | 0.51 | -23.43 |
| Fredlund & Xing | 0.51 | -63.15 | 2.43 | 47.13 |
| Hanson | 0.41 | -86.07 | 0.34 | -38.98 |
| van Genuchten | 0.50 | -65.53 | 0.36 | -36.61 |

The ELMv0 default run significantly underestimated both annual SR and GPP (Fig. 2). In addition, the simulated SR had smaller interannual variability compared to the observations. The model was not able to simulate the steep drop of SR or GPP during the extreme drought in 2012. The simulations of ST and SWP were isolated to analyse their contributions to model performance. Whereas the model simulated ST well at 10 cm depth (Fig. 3a), it tended to underestimate SWP when

water is limiting and to overestimate SWP otherwise (Fig. 3b). Implementing the data-constrained Hanson model significantly improved the simulation of SWP, showing a greater $R^2$ and a much smaller RMSE than that of the default run (Fig. 3b). After improving the simulation of SWP, the model better matched the observed annual SR and GPP (Fig. 2). The mean annual simulations of SR and GPP fell into the 1 sigma (i.e., standard deviation) of observations (inserted plot in Fig. 2). The changes in annual SR and GPP (i.e., the differences between before and after the improved SWP simulation using the Hanson model) showed a linear relationship (Fig. S1). In addition, the improved soil water scheme using the Hanson model increased both the moisture modifiers of GPP and heterotrophic respiration (i.e., btran and $\xi_w$) during the peak growing season, and reduced $\xi_w$ during the non-growing season (Fig. S2). The btran is the transpiration beta factor, which controls the soil water limitation to transpiration and photosynthesis, while $\xi_w$ is the soil moisture modifier for heterotrophic respiration as shown in Eq. (9). While SOC when simulated by the model with different soil water schemes generally fell within the wide range of observations, the improved SWP simulations using the Hanson model increased SOC stocks (Fig. S3).

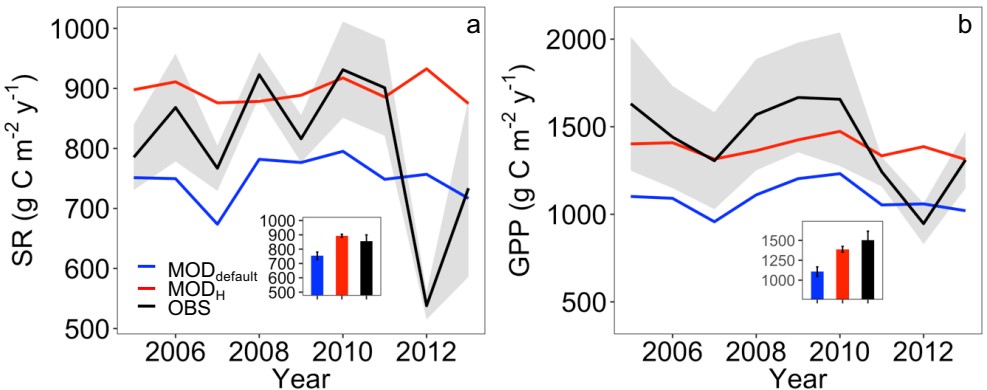

**Figure 2: Annual soil respiration (SR) and gross primary production (GPP).** Blue and red lines are model outputs before (MOD$_{default}$) and after (MOD$_H$) soil water potential improvement, respectively. Black lines and grey area are the observed (OBS) mean and 1 sigma (i.e., standard deviation) range, which were calculated from eight field replications for SR, and from three different net ecosystem exchange partitioning methods for GPP. The inserted bar plots are mean annual average ± 1 sigma across 2005-2011.

Despite the improved simulation of SR, the model still underestimated SR and GPP during peak growing seasons when SR and GPP were high, and overestimated them during non-growing seasons (Figs. 4, S4). In other words, though the improved simulation of SWP increased SR and GPP during peak growing seasons, the model still showed systematic errors. We attempted to improve the seasonal simulations of SR, GPP and LAI by modifying several related parameters (Table 2). Using measurements of C and energy fluxes from the MOFLUX site, Lu et al. (2018) calibrated a polynomial surrogate model of the ELMv0. Based on their results, we modified two parameters, i.e., the SLA at the canopy top from 0.03 to 0.01, and the fraction of leaf nitrogen in the RuBisCO enzyme from 0.1007 to 0.12.

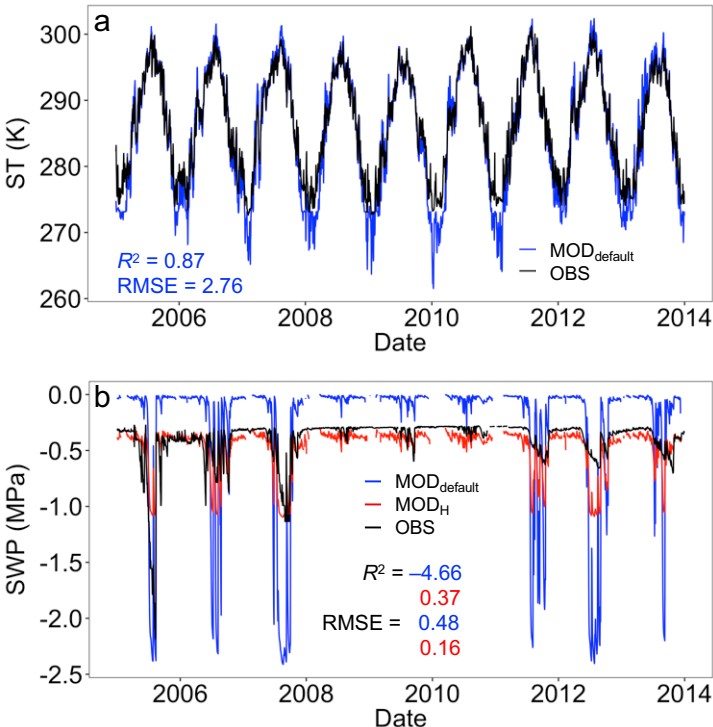

**Figure 3: Daily soil temperature (ST) and soil water potential (SWP) at 10 cm.** Blue and red lines/dots are model outputs before (MOD_default) and after (MOD_H) soil water potential improvement, respectively. $R^2$ and RMSE are shown in corresponding colours. Extremely low SWP values due to frozen soil water are not shown.

Comparing the simulated LAI with the observations (Fig. 4), we found that the parameter *ndays_off* (number of days to complete leaf offset) in the ELMv0 was too short (default value = 15 days) for the MOFLUX site. Thus, we reset the value of *ndays_off* to 45 days. We also modified the values of two additional parameters, i.e., *crit_dayl* and $\alpha$ correspondingly (Table 2). Parameter *crit_dayl* (the critical day length for senescence, units: second) triggers the leaf falling during the end of the growing season. Parameter ($\alpha$) is used to produce a linearly-increasing litterfall rate. Results showed that the ELMv0 with both the default and improved SWP by the Hanson model overestimated the maximum LAI (Fig. 4a). The adjustment of the aforementioned five parameters (Table 2) significantly reduced the LAI to within a more reasonable range (Fig. 4a). The parameter changes further increased the simulated GPP and SR during the peak growing season, in addition to the improvement by the adjusted SWP (Fig. 4b, c). However, all modifications of the ELMv0 still overestimated SR during the non-growing season, resulting in significant overestimation of annual SR fluxes (Fig. S5a). After the parameter adjustments, the annual GPP flux was still within the observed range (Fig. S5b). The contributions of autotrophic and heterotrophic

respiration to total SR had a seasonal cycle (Fig. 5). The contribution of heterotrophic respiration to total SR ranged from 60% to 90%.

**Table 2.** Modified parameters to better simulate gross primary production (GPP) and leaf area index (LAI) at the MOFLUX site in the ELMv0.

| Parameter name (unit*) | Parameter description | Default model value | Tuned values |
|---|---|---|---|
| *slatop* | Specific leaf area at top of canopy | 0.03 | 0.01 |
| *flnr* | Fraction of leaf nitrogen in RuBisCO enzyme | 0.1007 | 0.12 |
| *ndays_off* (d) | Number of days to complete leaf offset | 15 | 45 |
| *Crit_dayl* (s) | Critical day length for senescence | 39300 | 43200 |
| $\alpha$ | To control the rate coefficient $r_{xfer\_off}$ to produce a linearly-increasing litterfall rate | 2 | 10 |

*slatop*, *flnr* and $\alpha$ are unitless

In addition, we analyzed changes in simulated evapotranspiration (ET), runoff, photosynthesis, net primary production, C allocations to fine roots, leaf and woody tissue in response to the changes in the soil water scheme and parameters (Fig. S6, S7). The change in soil moisture scheme and parameter adjustments slightly increased ET and decreased runoff. Despite these slight changes, the model simulated ET generally fell within the observed range, with or without changes in soil water scheme and parameters (Fig. S6). The improved SWP and parameter adjustments generally increased all photosynthesis, NPP and carbon allocations to different tissues during the growing season (Fig. S7).

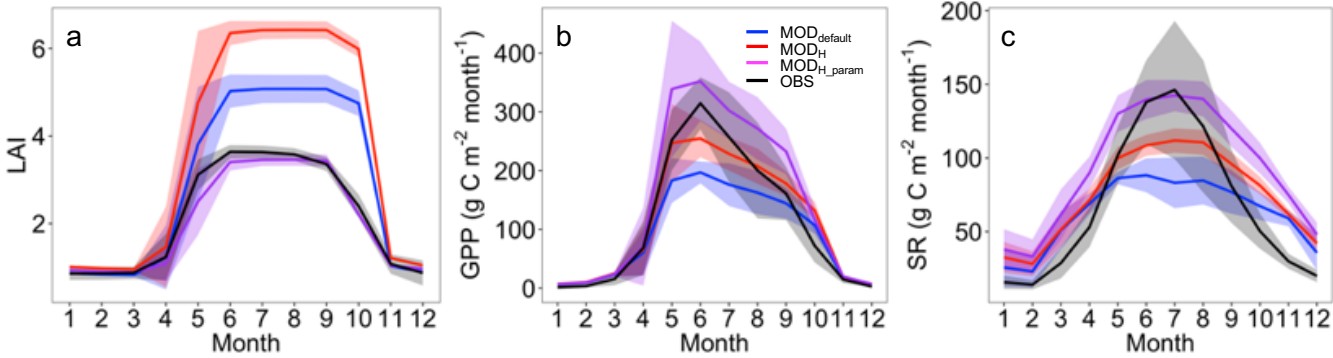

**Figure 4 The annual mean cycles of leaf area index (LAI), gross primary production (GPP) and soil respiration (SR).** OBS: observation; MOD_default: model output before soil water potential improvement; MOD_H: model output after soil water potential

improvement by the Hanson model; MOD$_{H\_param}$: model output after soil water potential improvement by the Hanson model and parameter adjustments.

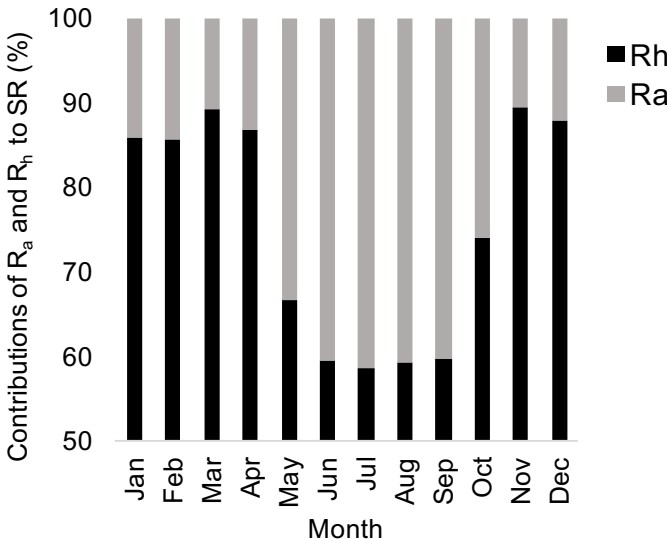

**Figure 5: Modelled contributions of autotrophic (R$_a$) and heterotrophic (R$_h$) respiration to total soil respiration (SR).**

## 4 Discussion

### 4.1 Effect of SWP on annual SR

Constraining the SWP-VWC relationship with site-specific data and using the Hanson model instead of the ELMv0 default model (Fig. 1) significantly improved the model representation of SWP (Fig. 3) and annual SR (Fig. 2a). The improvements in model fits could be due to the following reasons. First, the changes in SWP with the Hanson model increased plant transpiration and GPP in the model. The default ELMv0 underestimated GPP (Fig. 2b), similar to a recent study where CLM4.5 significantly underestimated GPP at a coniferous forest in northeastern United States (Duarte et al., 2017). GPP can directly affect the magnitude of root respiration as shown in many previous studies (Craine et al., 1999; Högberg et al., 2001; Wan and Luo, 2003; Verburg et al., 2004; Gu et al., 2008). Additionally, increased GPP can build a larger SOC pool which is the substrate for heterotrophic respiration (Fig. S3). Second, the Hanson soil moisture model increased the moisture modifier ($\xi_W$) on heterotrophic respiration during the peak growing season, and decreased it during the non-growing season

(Fig. S2), which is consistent with the trend of changes in SWP (Fig. 3). These changes together resulted in the improvement of simulated SR. In addition, the improvement of GPP and SR simulations was primarily due to the better simulation of the SWP in the upper 30 cm of the soil, as approximately 60% of plant roots are distributed in the upper 30 cm of the soil in temperate forests (Jackson et al., 1996). One important trend at the MOFLUX site was that soil moisture was lower during the peak growing season than during other times. As a result, the improved SWP simulation in the upper 30-cm soil during the peak growing season played a critical role in the improved simulation of GPP and SR.

The simulation of SWP in the default ELMv0 was poor compared with that of ST (Fig. 2), which may be a common issue in ESMs. For example, using a reduced-complexity model, Todd-Brown et al. (2013) demonstrated that the spatial variation in soil C in most ESMs is primarily dependent on C input (i.e., NPP) and ST, showing $R^2$ values between 0.62 and 0.93 for 9 of 11 ESMs. However, the same reduced-complexity model, driven by observed NPP and ST, can only explain 10% of the variation in the Harmonized World Soil Database observational database (Todd-Brown et al., 2013). These previous results indicate that other important factors affecting soil C dynamics, in addition to NPP and ST, are inadequately simulated in ESMs (Powell et al., 2013; Reyes et al., 2017). Powell et al. (2013) showed that differential sensitivity of SR to VWC in several ESMs using observations in two Amazon forests. Our analyses in this study indicate that improving the modelled SWP can significantly improve mean annual GPP and SR simulations. Thus, we propose that the SWP simulation in ESMs should be calibrated carefully with observations, and/or by using different model representations of the SWP-VWC relationship. Because there is no global grid-based SWP database, paired measurements of VWC and SWP are needed along with soil characteristics in a variety of soil types and ecosystems. These data can be used to calibrate SWP-VWC relationships and SWP simulations in models. Besides, there are many sites, such as the MOFLUX site in this study, collecting long-term hydrological and biogeochemical data. These data are useful to evaluate whether better SWP simulation will improve biogeochemical cycling simulations.

In this study, we derived a better SWP-VWC relationship by using non-linear fitting, primarily because of the availability of soil moisture retention curve data. It is an efficient method when site-level data is available, but it is not realistic to calibrate the water retention curve for every site. The SWP-VWC relationship is dependent on soil texture (Clapp and Hornberger, 1978; Cosby et al., 1984; Tuller and Or, 2004), so building relationships between model parameters and soil texture may allow efficient extrapolations of site-level measurements to regional and global scales.

Parameters in the default Clapp & Hornberger model used in the ELMv0 were derived from synthesizing data across soil textural classes (Clapp and Hornberger, 1978; Cosby et al., 1984; Lawrence and Slater, 2008). The data were derived from over 1,000 soil samples from 11 USDA soil textural classes (Holtan et al., 1968; Rawls et al., 1976). The dependence of model parameters on soil texture were derived from a regression of these 11 data points, i.e., the mean parameter values of 11 soil textural classes against the sand or clay fractions (Cosby et al., 1984). Because no actual sand or clay content of soil samples was reported in the original databases (i.e., only the soil textural classes were reported), the sand and clay fractions used for the regression were obtained from midpoint values of each textural class (Clapp and Hornberger, 1978; Cosby et al., 1984). One potential issue is that soil samples in the same textural classes can have different sand and clay contents and SWP-VWC relationships, which may not be fully represented when they are grouped together. An updated SWP-VWC

database with actual sand and clay content measurements could provide improved empirical relationships between model parameters and soil texture in the water retention model.

In addition, different empirical models have been developed to describe the SWP-VWC relationship (Brooks and Corey, 1964; Clapp and Hornberger, 1978; van Genuchten, 1980; Fredlund and Xing, 1994; Hanson et al., 2003). These models could be evaluated against data, and the selected best-fit model(s) could be used to calculate SWP in the field from continuously monitored VWC (e.g., from the AmeriFlux network) on different spatial and temporal scales. The database could also be used as a benchmark to evaluate simulations of soil water and biogeochemical processes in ESMs.

Moreover, we also explored whether the calibrated Clapp & Hornberger model can lead to similar improvements with the Hanson model (Fig. S8). Generally, both the Hanson model and the calibrated Clapp & Hornberger model improved the simulation of GPP and SR in the ELM, in comparison with the default run (Fig. S8). The ELMv0 with the Hanson model consistently produced higher GPP and SR than that with the calibrated Clapp & Hornberger model. In comparison with the observations, the modelled SR generally fell within the 1 sigma (i.e., standard deviation) range of observations, by using both the Hanson model and the calibrated Clapp & Hornberger model. However, the modelled GPP with the calibrated Clapp & Hornberger model was still lower than the observations. Given the order of the goodness-of-fit of the SWP-VWC relationship was default Clapp & Hornberger model < calibrated Clapp & Hornberger model < calibrated Hanson model (Table 1), these results further support the conclusion that better representations of SWP can improve the simulations of carbon dynamics. Therefore, throughout the remainder of this manuscript, we used the Hanson model to represent the SWP-VWC relationship.

## 4.2 Representation of seasonal and interannual variabilities in the ELMv0

Although the SWP simulations using the Hanson model improved the representation of both annual SR and GPP, the model continued to overestimate SR during the non-growing season (Figs. 4), resulting in significant overestimations of the annual SR fluxes (Fig. S5). No matter which SWP simulations were used, the ELMv0 had smaller interannual variability than the observations (Fig. 2). Specifically, the model was not able to capture the steep decreases in GPP and SR in the extreme drought year (i.e., 2012; Fig. S9). These results indicate that the current model structure is not sensitive enough to environmental changes. Several potential reasons may contribute to the underestimated seasonal and interannual variability. For example, field inventory data at the study site showed that the severe drought-pathogen interactions in 2012 resulted in a significant stem mortality of tree species (Wood et al., 2017). Thus, the observed steep decreases in GPP and SR could be due to mortality. The stem mortality could lead to lower evapotranspiration (Fig. S9), minimizing soil moisture losses (Fig. S10). However, the ELMv0 simulated the moisture effect on biogeochemical cycles at the physiological level, but not at the plant community level. In addition, the strong dependence of GPP and SR on the upper layer soil moisture could explain the model's difficulty in capturing inter-annual variability. Although better representation of SWP improved the mean annual simulation of biogeochemical processes, the model could not capture the mortality or the interannual variability of GPP and SR.

The calculation of the moisture scalars (e.g., btran and $\xi_W$) using empirical equations from SWP may be another potential reason for the insensitivity. For example, observational results have shown that there may be an optimal moisture point at which soil respiration peaks with significant reductions in decomposition towards both dryer and wetter conditions (Linn and Doran, 1984; Franzluebbers, 1999; Monard et al., 2012; Sierra et al., 2017). In the ELMv0, however, the moisture scalar increases from 0 to 1 with the increase in soil moisture and does not decrease afterwards (Eq. 9). Thus, the ELMv0 may not be sensitive to extreme wet conditions. The linear empirical equation between the lower and upper thresholds ($\Psi_{min}$ and $\Psi_{max}$) may not capture non-linear moisture behaviours, leading to insensitive responses of biogeochemical processes to moisture change. Incorporating more mechanistic moisture scalars may improve the sensitivity of the model in response to moisture changes (Ghezzehei et al.; Yan et al., 2018).

In the ELMv0, heterotrophic respiration contributed the majority (i.e., over 85%) of total SR during non-growing seasons (Fig. 5), suggesting that the overestimation of SR during these seasons was primarily due to the biased heterotrophic respiration simulation. A potential reason for the biased heterotrophic respiration simulation may be related to the temperature sensitivity ($Q_{10}$). Theoretically, a higher $Q_{10}$ can result in greater seasonal variability of SR (Fig. S11). Compared to relatively small $Q_{10}$ values, a larger $Q_{10}$ can lead to lower heterotrophic respiration when temperature is below the reference temperature, and greater heterotrophic respiration when temperature is above the reference (Fig. S11). In the ELMv0, the reference temperature is 25 ºC and the $Q_{10}$ of heterotrophic respiration is 1.5 (Oleson et al., 2013). A previous study derived a much greater $Q_{10}$ value (i.e., 2.83) when the parameters were calibrated with data from another temperate forest (Mao et al., 2016). We hypothesized that the $Q_{10}$ value of 1.5 may be too small for the MOFLUX site. We arbitrarily increased $Q_{10}$ from 1.5 to 2.5, but there were minimal effects on the SR simulation (Fig. S12). This indicates that modifying the temperature sensitivity of heterotrophic respiration may not improve the modelled representation of seasonality of SR in the ELMv0.

Another potential reason for the biased heterotrophic respiration simulation may be that the seasonality of microbial organisms was not adequately represented in the model. Like most ESMs, the ELMv0 represents soil C dynamics using linear differential equations and assumes that SR is a substrate-limited process in the model. However, producers of $CO_2$ in soils, microbial organisms, have a significant seasonal cycle (Lennon and Jones, 2011). These organisms usually have very high biomass and activity during growing season peaks with favourable conditions of temperature, moisture and substrate supply, and tend to be dormant under stressful conditions (Lennon and Jones, 2011; Stolpovsky et al., 2011; Wang et al., 2014; Wang et al., 2015). The seasonality of microbial biomass and activity, in addition to that of GPP and ST, may contribute to the seasonal variability of SR.

Additionally, the lack of representation of macroinvertebrate and other forest floor and soil fauna in the ELMv0 may be another reason. There is a high density of earthworms at the MOFLUX site (Wenk et al., 2016). Earthworms can shred and redistribute soil C and change soil aggregation structure, which may alter soil C dynamics and $CO_2$ efflux to the atmosphere (Verhoef and Brussaard, 1990; Brussaard et al., 2007; Coleman, 2008). Like microbial organisms, earthworms usually have a significant seasonal cycle, showing high biomass and high activity during peak growing seasons and tending to be dormant during non-growing seasons (Wenk et al., 2016). However, a recent review suggests that current experimental evidence and

conceptual understanding remains insufficient to support the development of explicit representation of fauna in ESMs (Grandy et al., 2016). Therefore, data collection focused on seasonal variations in fauna and microbial biomass and activity might enable further improvements in the representation of seasonal variation in SR.

Our analyses also showed that the modelled SR was not able to reach the observed peak in many years during the peak growing season, even when the modelled GPP exceeded the observation. In addition, the parameter modification increased GPP during both peak and non-growing seasons, resulting in an even greater overestimation of SR during non-growing seasons. These results suggest that simply increasing GPP may not be adequate to increase the seasonal variability of the simulated SR. A potential reason may be that the current model does not include root exudates. Root exudates are labile C substrates that are important for SR (Kelting et al., 1998; Kuzyakov, 2002; Sun et al., 2017). The root exudate rate is primarily dependent on root growth, showing a seasonal cycle in temperate forests (Kelting et al., 1998; Kuzyakov, 2002). Thus, including root exudates in the model may further increase the model simulated SR during the peak growing season without needing to increase GPP.

## 5 Conclusions

In this study, we used temporally extensive and spatially distributed site observations of SR to assess the capabilities of ELMv0. These results indicated that an improved representation of SWP within the model provided better simulations of annual SR. This underscores the need to calibrate SWP in ESMs for more accurate projections of coupled climate and biogeochemical cycles. Notwithstanding this improvement, however, the ELMv0 still underestimated seasonal and interannual variabilities. It may be that inadequate model representation of vegetation dynamics, moisture function, and the dynamics of microbial organisms and soil macroinvertebrates could be explored as means to achieve better fit. Future incorporation of explicit microbial processes with relevant data collection activities may therefore enable improved model simulations.

*Code availability*. The code for ELMv0 is available at https://e3sm.org.

*Data availability*. The data for this paper are available upon request to the corresponding author.

*Competing interests*. The authors declare that they have no conflict of interest.

*Author contribution*. JL, GW and MAM designed the study. JL, GW and DMR ran the model. LG, PJH and JDW contributed to data collection. JL wrote the paper with input from all authors.

*Acknowledgements*. Authors thank Dan Lu for sharing unpublished data and constructive comments. This work is financially supported by the U.S. Department of Energy (DOE) Office of Biological and Environmental Research through the Terrestrial Ecosystem Science Scientific Focus Area (TES-SFA) at Oak Ridge National Laboratory (ORNL), the Climate Model Development and Validation (CMDV) project, and the Energy Exascale Earth System Model (E3SM) project. ORNL is managed by UT-Battelle, LLC, under contract DE-AC05-00OR22725 with the U.S. DOE.

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
