# Peer review of "Figure S1. Residual (modeled – observed) of soil respiration (SR) from 2005 to 2013. The"

_Geoscientific Model Development, 2018_

## Short Comment (SC1) · 20 Mar 2018

Dear authors,

in my role as Executive editor of GMD, I would like to bring to your attention our Editorial version 1.1:

http://www.geosci-model-dev.net/8/3487/2015/gmd-8-3487-2015.html

This highlights some requirements of papers published in GMD, which is also available on the GMD website in the 'Manuscript Types' section:

http://www.geoscientific-model-development.net/submission/manuscript_types.html

[Figure]

In particular, please note that for your paper, the following requirement has not been met in the Discussions paper:

- "The main paper must give the model name and version number (or other unique identifier) in the title."

Even if this is not a strict requirement for evaluation paper, we like to encourage authors to provide also the version number of the evaluated model, as usually evaluation results depend on model version.

Additionally, please not, that GMD is encouraging authors to provide a persistent access to the exact version of the source code used for the model version presented in the paper. As explained in https://www.geoscientific-model-development.net/about/manuscript_types.html the preferred reference to this release is through the use of a DOI which then can be cited in the paper. For projects in GitHub (such as thee E3SM Land Model) a DOI for a released code version can easily be created using Zenodo, see https://guides.github.com/activities/citable-code/ for details. Yours,

Astrid Kerkweg

---

## Referee Comment (RC1) · W. Wieder (Referee) · 3 Apr 2018

Liang and coauthors present a nice study exploring sensitivities in the E3SM land model to changes in the calculation of soil water potential and subsequently to parametric changes related to plant physiology. They compare simulated results to observations from the MOFLUX site, focusing on carbon fluxes (GPP and soil respiration, SR).

There are a host of changes suggested that generally improve agreement with observed results, but in general it's hard to follow what changes are most important for the improvements. I appreciate the need to keep text and display items simple & digestible for readers, but a bit more complexity would help shed light on the factors responsible for the site-level improvements in the model made here.

For example, it looks like the modified soil water potential scheme (Hanson, I think), provides a better fit to GPP, SR and soil moisture (Figs. 2-5), but it remains unclear if the modifications are significantly better (or different) from the Clapp & Hornberger scheme that's been tuned to local edaphic characteristics? It's not that surprising that the parameterization for a global model would not be a good fit to local results, so does the model just need tuning for site-level runs, or are underlying physics and assumptions in the Hanson scheme fundamentally superior to another approach? Addressing this question matters if the long-term aim of this work is to document changes made to ELM from CLM4.5.

Similarly, how important are the suggested parameter changes for capturing the annual cycle of LAI and C fluxes vs. changes to the soil moisture scheme (Fig. 6). Stepping through these changes sequentially in the text and display items will clarify the source(s) of the improvements.

Finally, although the authors claim that improving SWP directly improved soil respiration estimates, it's not clear if this is a direct effect of soil moisture on soil respiration, or merely reflective of the larger plant and soil C stocks simulated as a result of having higher GPP. Concurrently presenting changes to ecosystem C stocks and the soil moisture effect on GPP and heterotrophic respiration (btran and w_scalar, respectively in CLM4.5) will help clarify how / why improvements were made.

Major concerns

I appreciate the effort used to explore alternative formulations for SWP in the model (Fig. 1, Table 1). Two questions come to mind. First, is it worth doing a more thorough model selection process like AIC or BIC that penalizes more complex models for their additional parameters instead of just showing RMSE. Second, as E3SM is intended to be run in global simulations, I wonder what effect alternative formulations for SWP

have on water and energy fluxes from the model in site level, and ultimately global, simulations? The GPP results (Fig. 2) are a good start for this, but presumably these changes really modify ET fluxes (and runoff). It seems documenting these changes are likely important (if only in SI)?

If Hanson or van Genuchten formulations are 'better' fits to the observations, why aren't they used for GPP simulations in Fig. 2? What's the purpose of exploring alternative SWP schemes, if they don't follow through the C cycle simulations in the model? Reading the text on the bottom of page 7, however, maybe (MODswp) is using the Hanson scheme? If so, does the calibrated Clapp & Hornberger approach provide similar improvement by removing the high bias in the default configuration (Fig. 3). Please clarify in the text and figure captions what's being shown and why none of the models adequately capture the effect of the 2012 drought.

Minor concerns

Section 2.3 This is really a broader comment on how author groups working with E3SM intend to articulate the version of the model on which they are working, esp. for readers less familiar with nuances of CLM4.5 development branches and subsequent ELM developments. For example, how is this code different from other publications (e.g. Brunke et al. 2016; Riley 2018)?

Section 2.3. Please justify the decision to use the CLM-CN decomposition module for a paper focused on soil respiration when Bonan and others (2013) clearly demonstrated shortcomings of this model version? It seems like the wrong tool for this job?

Section 2.3. This is also a little confusing, as the opening line of the section states the soil biogeochemistry is vertically resolved, but to my knowledge CLM-CN does not apply vertically resolved soil BGC? Please clarify

Page 4, Line 20. Single point runs (especially with CLM) forced with flux tower measurements have a long history that should be acknowledged here.

Page 7, Line 7. What changes were made to the Clapp and Hornberger parameterization, there are lots of hard coded parameteris in eq. 11-13.

Fig 2. Why are observations shown with a black line and purple bar (inset)? Consistency within and among figures will help readers understand display items more easily? Similarly, using the same color for line of the default model and modified model in Figs. 1 and 2 would be helpful

One strength of using flux tower data in single point simulations seems to be examining the seasonal cycle of carbon and energy fluxes. This is somewhat lost in Fig. 6, and I wonder if the display item would be more powerful if simulations are results were averaged over the whole observation record (e.g. just show 1 year instead of 9, as the interannual variability isn't that obvious (and already shown in Fig. 2)

Fig 4 is never really discussed and doesn't add much to the paper in my estimation. Can it be removed from the text? More, it follows that that changes in productivity would have a linear effect on soil C stocks and therefore respiration rates in a first order model like CLM-cn (Todd-Brown refs from the text), so the relationship shown here isn't really surprising.

Out of curiosity, how do simulated soil (or vegetation) C stocks compare with observed stocks at the site? The focus on fluxes is fine, but given that fluxes are linear related with stocks, do suggested modifications to the model improve estimates of fluxes AND stocks for the site?

Seasonal biases in SR and GPP fluxes look pretty bad with default and 'swp' versions of the model (Fig. 5). The parametric changes in Table 2 seem to address some of these seasonal biases (Fig. 6), but it seems like showing the scatter plots on Fig. 5 (maybe with a 3rd color) would be helpful? Along these lines, should both Figs. 5 and 6 show the same 3 simulation ('default', 'swp', and 'swp_param')? Showing the mean annual cycle (+/- 1 sigma on the observations) for all panels in Fig. 6 would help to make this figure easier to digest.

SLA is something that's measured, maybe not at the site for similar trees to the ones at the site? Is building 3x thicker leaves (Table 2), a reasonable assumption? Similarly, if the authors need to decrease LAI while increasing GPP, flnr necessarily has to increase in the model, but is the 20% increase here supported by databases like TRY, or are these parameter changes just illustrating big nobs in the model that are poorly constrained by observaions?

Page 10, line 12 please report statistics to support claims being made. Visually, the red line looks closer to the observations than the blue one (Fig. 6b,c). How do the annual totals look?

As with comment above, how do changes in annual fluxes or total stocks compare with observations following parametric changes suggested in Table 2?

Page 12, line 4. This doesn't seem like a fair statement or comparison, as results from the tuned Clapp & Hornberger scheme are never presented.

Page 12, line 10, given the dominance of Rh in contributions to soil respiration (Fig 7). I'd suspect that changes in SR have more to do with larger SOM stocks than they do links between substrate supply through GPP, as suggested here, but no data are presented along these lines?

Page 12, line 20. This statement may be true, but it's not clear that changes to VMC proposed here had much of an effect on the Rh component of the model. To show this, it seems like showing the soil moisture effect (w_scalar) on soil decomposition rates from different configurations of the model would be needed. Otherwise, I'd suspect that improvement to SR (Fig 2, 5) are predominantly driven by larger soil C stocks (via higher GPP), but not from direct improvement in the SWC on soil biogeochemistry, as suggested in the Powell paper referenced.

Page 12, line 20 what is SWP-VMC, should be VWC?

Page 13, line 2. Again this claim is poorly supported by the data presented. Yes, the

tuned Clapp and Hornberger model is not the 'best' model in Table 1, but are results for GPP or SR markedly different than the Hansen results shown?

Page 14. The q10 analysis is nice, but I wonder if a more ecological explanation is relevant here- specifically highlighting the role of root exudates in supplying labile C substrates that are important for SR? The land model here doesn't consider these ecologically important C fluxes that likely have an important control over the seasonal dynamics of soil respiration and microbial biomass already discussed?

References: Bonan, et al (2013) Evaluating litter decomposition in earth system models with long-term litterbag experiments: an example using the Community Land Model version 4 (CLM4), Global Change Biology, 19, 957-974.

Brunke et al. 2016. "Implementing and Evaluating Variable Soil Thickness in the Community Land Model, Version 4.5 (CLM4.5)." Journal of Climate 29(9): 3441-3461, doi:10.1175/JCLI-D-15-0307.1.

Riley, W. 2018. "Impacts of Microtopographic Snow Redistribution and Lateral Subsurface Processes on Hydrologic and Thermal States in an Arctic Polygonal Ground Ecosystem: A Case Study Using ELM-3D v1.0." Geoscientific Model Development 11(1): 61-76, doi:10.5194/gmd-11-61-2018.

---

## Referee Comment (RC2) · Anonymous Referee #2 · 13 Apr 2018

This paper reports an effort of tuning an Earth system model, E3SM, to fit observed leaf area index (LAI), gross primary production (GPP, derived from eddy flux data), and soil respiration at a temperate deciduous forest site. The authors specifically tested different empirical relationships between volumetric water content (VWC) and soil water potential (SWP), and found tuning soil water potential improve the simulation of soil respiration. So, they concluded that "modelling soil respiration can be significantly improved by better model representations of the soil water retention curve." I agree with the authors that the well data-constrained model, Hanson model, increased the prediction of soil water potential, and may improve the simulation of GPP, which have been shown by the results (Figs. 3 and 7). But for the improvement of soil respiration, I

think it's just a coincidence. From the Fig. 5a (page 9), we can see the new VMC-SWP relationship (i.e., Hanson model) increases soil respiration rate overall, but it does NOT change the pattern. This means the performance of soil respiration modeling is not improved. The authors also pointed out that the original model underestimates GPP and soil respiration (Line 13, page 7, and Fig. 2). So, the improvement of soil respiration prediction was not due to the improvement of SWP simulation, but because increases in GPP. The increases in GPP may increase carbon allocation to roots or total soil carbon, and therefore increase soil respiration. And, according to Fig. 7, the most possible reason for underestimating soil respiration is that the root respiration is not high enough in growing season, which also leads to the seasonal pattern that does not fit the observations because root respiration is usually high in growing season and very low in non-growing season.

A detailed report on the tuning of an ESM is valuable even if no new mechanisms were added. It helps to understand model performance and the thoughts behind the model development. For improving simulation of soil respiration, the authors had looked at the sensitivity to temperature, LAI, GPP, and relative contributions of roots and soil carbon, and tuned a bunch of parameters (Table 2 in page 5). A detailed analysis of the successes and fails of these tunings would be interesting. For example, I'd like to see how the improvement of SWP prediction affects plant physiology, photosynthesis, allocation, NPP (because NPP=Rh at equilibrium). These variables may change soil respiration.

Specifically, for water effects on soil heterotrophic respiration, the model uses two equations to link volumetric water content to heterotrophic respiration: VMC–>SWP and SWPRh. The second equation (SWPRh, Eq 9 in page 4) is much more critical than the first one for modeling heterotrophic respiration. It represents the knowledge of how soil moisture affects microbial physiology. It needs to be explored in detail if the goal of this research is to improve the simulation of soil respiration.

---

## Author Comment (AC1) · 29 May 2018

**Letter of Responses**

Authors' note: The original reviewers' comments are *in italic and colored blue*, and our responses follow. All page/line numbers indicated in the responses are those in the revision.

*Reviewer 1 (Dr. W. Wieder)*

*Liang and coauthors present a nice study exploring sensitivities in the E3SM land model to changes in the calculation of soil water potential and subsequently to parametric changes related to plant physiology. They compare simulated results to observations from the MOFLUX site, focusing on carbon fluxes (GPP and soil respiration, SR).*

*There are a host of changes suggested that generally improve agreement with observed results, but in general it's hard to follow what changes are most important for the improvements. I appreciate the need to keep text and display items simple & digestible for readers, but a bit more complexity would help shed light on the factors responsible for the site-level improvements in the model made here. For example, it looks like the modified soil water potential scheme (Hanson, I think), provides a better fit to GPP, SR and soil moisture (Figs. 2-5), but it remains unclear if the modifications are significantly better (or different) from the Clapp & Hornberger scheme that's been tuned to local edaphic characteristics? It's not that surprising that the parameterization for a global model would not be a good fit to local results, so does the model just need tuning for site-level runs, or are underlying physics and assumptions in the Hanson scheme fundamentally superior to another approach? Addressing this question matters if the long-term aim of this work is to document changes made to ELM from CLM4.5.*

**Response:** We greatly appreciate the valuable comment. Particularly interesting is the question *"if the modifications are significantly better (or different) from the Clapp & Hornberger scheme that's been tuned to local edaphic characteristics"*. In the revised manuscript, we compared the simulated gross primary production (GPP) and soil respiration (SR) when using the Hanson model and the calibrated Clapp & Hornberger model. Generally, both the Hanson model and the calibrated Clapp & Hornberger model improved the simulation of GPP and SR in the E3SM Land Model version 0 (ELMv0), in comparison with the default run with the uncalibrated Clapp & Hornberger model (Fig. S8 and also see below). The ELMv0 with the Hanson model consistently produced higher GPP and SR than that with the calibrated Clapp & Hornberger model. In comparison with the observations, the modeled SR generally fell within the 1 sigma (i.e., standard deviation) range of observations, by using both the Hanson model and the calibrated Clapp & Hornberger model. However, the modeled GPP with the calibrated Clapp & Hornberger model was lower than the observations. Given the goodness-of-fit of the soil water potential (SWP)-volumetric water content (VWC) relationship was default Clapp & Hornberger model < calibrated Clapp & Hornberger model < calibrated Hanson model (revised Table 1 and also see below), these new results further support our conclusion that better representations of SWP can improve the simulations of carbon processes (i.e., GPP and SR here).

In the revised manuscript, we added a new supplementary figure (i.e., Fig. S8) and a paragraph in the text to compare simulations with the Hanson model and the calibrated Clapp & Hornberger model (page 12, lines 10 – 19):

*"Moreover, we also explored whether the calibrated Clapp & Hornberger model can lead to similar improvements with the Hanson model (Fig. S8). Generally, both the Hanson model and the calibrated Clapp & Hornberger model improved the simulation of GPP and SR in the ELM, in comparison with the default run (Fig. S8). The ELMv0 with the Hanson model consistently produced higher GPP and SR than that with the calibrated Clapp & Hornberger model. In comparison with the observations, the modeled SR generally fell within the 1 sigma (i.e., standard deviation) range of observations, by using both the Hanson model and the calibrated Clapp & Hornberger model. However, the modeled GPP with the calibrated Clapp & Hornberger model was still lower than the observations. Given the goodness-of-fit of the soil water potential (SWP)-volumetric water content (VWC) relationship was default Clapp & Hornberger model < calibrated Clapp & Hornberger model < calibrated Hanson model (Table 1), these results further support the conclusion that better representations of SWP can improve the simulations of carbon processes. Therefore, throughout the remainder of this manuscript, we used the Hanson model to represent the SWP-VWC relationship."*

[Figure]

**Figure S8: Annual soil respiration (SR) and gross primary production (GPP).** Blue lines are the ELMv0 simulations with default parameters ($MOD_{default}$), red lines with the soil water potential improved using the calibrated Clapp & Hornberger model ($MOD_{cCP}$), and purple lines with the soil water potential improved using the Hanson model ($MOD_H$). Black lines and grey area are the observed (OBS) mean and 1 sigma range, which were calculated from 8 field replications for SR, and from three different net ecosystem exchange partitioning methods for GPP. The inserted bar plots are mean annual average ± 1 sigma across 2005-2011.

The reviewer also asked *"are underlying physics and assumptions in the Hanson scheme fundamentally superior to another approach"*. The short answer is no. Although different approaches for soil water retention curve may have different underlying physics and assumptions, pragmatic models are those which have been well calibrated/parameterized with

empirical data. Since the default ELMv0 simulated the SWP poorly at the MOFLUX site (Fig. 3b), one important question we asked in this study was whether better representation of SWP in the model would improve the simulations of carbon processes. To improve the SWP simulation as much as possible, our effort was not limited to tuning the Clapp & Hornberger model since we did not know whether the tuned Clapp & Hornberger model would be good enough to answer the question. Instead, we evaluated a series of soil water retention curve models popularly used in the literature to derive the best-fit model using root-mean-square-error (RMSE) and Akaike Information Criterion (AIC) as suggested by the reviewer in another comment (revised Table 1 and also see below). The Hanson model performed the best, showing the smallest RMSE and AIC values. Both the modeled annual fluxes of GPP and SR fell within the 1 sigma range of observations when using the Hanson model, but not with the calibrated Clapp & Hornberger model as shown in Fig. S8 and above. Thus, we used the Hanson model for all further analyses. In the reminder of this response letter and the manuscript, the improved SWP was simulated using the Hanson model if not otherwise specified, and all changed ELMv0 simulations were compared to the default simulations with the default Clapp & Hornberger model.

**Table 1.** Root-mean-square-error (RMSE) and Akaike Information Criterion (AIC) of different models in simulating the SWP-VWC relationship for the soil in the MOFLUX site at two depths: 0 to 30 cm and below 30 cm.

| Model | < 30 cm | | > 30 cm | |
|---|---|---|---|---|
| | RMSE | AIC | RMSE | AIC |
| Clapp & Hornberger (default ELMv0) | 4.25 | 157.82 | 1.33 | 18.51 |
| Brooks & Corey | 3.91 | 151.05 | 1.13 | 13.51 |
| Clapp & Hornberger (calibrated) | 0.53 | -61.03 | 0.51 | -23.43 |
| Fredlund & Xing | 0.51 | -63.15 | 2.43 | 47.13 |
| Hanson | 0.41 | -86.07 | 0.34 | -38.98 |
| van Genuchten | 0.50 | -65.53 | 0.36 | -36.61 |

Although it needs further exploration as to whether the Hanson model performs the best on the regional and global scales, the default Clapp & Hornberger model used in the ELMv0 performs poorly in simulating SWP on the global scale (See below), which may significantly impact the biogeochemical simulations. In a different (but related) project, we tested the simulated SWP by the default Clapp & Hornberger model used in the ELMv0 against 6928 data points of paired measurements of SWP and VWC across different soil types and ecosystems. Results showed that the default Clapp & Hornberger model used in the ELMv0 was not able to reproduce the observed SWP (see Fig. R1 below). It remains unclear which model will perform best in describing the SWP-VWC relationship on the global scale. More work will be needed to explore the issue, which is beyond the scope of this manuscript.

[Figure]

**Figure R1: Comparison of observed and simulated soil water potential (-MPa) across different soil types and ecosystems by the default Clapp & Hornberger model in the ELMv0.**

*Similarly, how important are the suggested parameter changes for capturing the annual cycle of LAI and C fluxes vs. changes to the soil moisture scheme (Fig. 6). Stepping through these changes sequentially in the text and display items will clarify the source(s) of the improvements.*

**Response:** The reviewer provided a great suggestion. Thanks to the reviewer's other suggestion, we are able to more clearly show the importance of the parameter changes and the improved SWP by plotting the mean annual cycle (± 1 sigma) of LAI, GPP and SR (revised Fig. 4 and also see below). With the revised figure, we can step through the changes. Results showed that the ELMv0 with both the default and improved SWP by the Hanson model overestimated the maximum LAI (Fig. 4a). The parameter changes significantly reduced the maximum LAI to better match the observations (Fig. 4a). The parameter changes further increased the simulated GPP and SR during the peak growing season, in addition to the improvement by the adjusted SWP (Fig. 4b, c). However, all modifications of the ELMv0 still overestimated SR during the non-peak growing season, as discussed (previously) in Section 4.2.

[Figure]

**Figure 4 The annual mean cycles of leaf area index (LAI), gross primary production (GPP) and soil respiration (SR).** OBS: observation; $MOD_{default}$: model output before soil water potential improvement; $MOD_H$: model output after soil water potential improvement by the Hanson model; $MOD_{H\_param}$: model output after soil water potential improvement by the Hanson model and parameter adjustments.

In the revised manuscript, we revised the text accordingly to show the effect of the parameter changes on the simulations of LAI, GPP and SR (page 10, line 6 – 10):

*"Results showed that the ELMv0 with both the default and improved SWP by the Hanson model overestimated the maximum LAI (Fig. 4a).The adjustment of the aforementioned five parameters (Table 2) significantly reduced the LAI to within a more reasonable range (Fig. 4a). The parameter changes further increased the simulated GPP and SR during the peak growing season, in addition to the improvement by the adjusted SWP (Fig. 4b, c). However, all modifications of the ELMv0 still overestimated SR during the non-peak growing season, resulting in significant overestimation of annual SR fluxes (Fig. S5a). After the parameter adjustments, the annual GPP flux was still within the observed range (Fig. S5b)."*

*Finally, although the authors claim that improving SWP directly improved soil respiration estimates, it's not clear if this is a direct effect of soil moisture on soil respiration, or merely reflective of the larger plant and soil C stocks simulated as a result of having higher GPP. Concurrently presenting changes to ecosystem C stocks and the soil moisture effect on GPP and heterotrophic respiration (btran and w_scalar, respectively in CLM4.5) will help clarify how / why improvements were made.*

**Response:** We appreciate the insightful comment. The second reviewer had a similar concern. We agree with the reviewers that concurrently presenting changes in ecosystem carbon stocks and the soil moisture effect on GPP (btran) and heterotrophic respiration ($\xi_w$ in equation 9) will help clarify how improvements were made. In the revised manuscript, we analyzed the changes in btran, $\xi_w$ and soil organic carbon (SOC) (Fig. S2-S3 and also see below). The improved soil water scheme using the Hansom model increased both btran and $\xi_w$ during the peak growing season, and reduced $\xi_w$ during the non-peak growing season (Fig. S2). The change in $\xi_w$ was generally consistent with that of SWP (Fig. 3b). While the model simulated SOC with different soil water schemes generally fell within the wide range of observations, the improved SWP simulations increased SOC stocks, matching the reviewer's expectation (Fig. S3). These results, combining with previous results (the original Fig. 4, which was moved as Fig. S1 in the revision as suggested by the reviewer's other comment), indicate that the improved soil respiration by SWP was a joint result of changes in GPP, SOC stocks and the moisture modifier of heterotrophic respiration.

In the revised manuscript, we added these new figures, presented the results and discussed details in the text.

*"The changes in annual SR and GPP (i.e., the differences between before and after the improved SWP simulation using the Hanson model) showed a linear relationship (Fig. S1). In addition, the improved soil water scheme using the Hanson model increased both the moisture modifiers of GPP and heterotrophic respiration (i.e., btran and $\xi_w$) during the peak growing season, and reduced $\xi_w$ during the non-peak growing season (Fig. S2). While SOC when simulated by the model with different soil water schemes generally fell within the wide range of observations, the improved SWP simulations using the Hanson model increased SOC stocks (Fig. S3)."*

*"Constraining the SWP-VWC relationship with site-specific data and using the Hanson model instead of the ELMv0 default model (Fig. 1) significantly improved the model representation of SWP (Fig. 3) and annual SR (Fig. 2a). The improvements in model fits were due to changes in GPP, SOC stocks, and the moisture modifier on heterotrophic respiration (Figs. S1 – S3). First, the default ELMv0 underestimated GPP (Fig. 2b), as in a recent study where CLM4.5 significantly underestimated GPP at a coniferous forest in northeastern United States (Duarte et al., 2017). GPP affects the substrate supply for SR, as evidenced by the close relationship between changes in SR and GPP (Fig. S1), which is consistent with experimental evidence showing GPP can directly affect the magnitude of root respiration (Craine et al., 1999; Högberg et al., 2001; Wan and Luo, 2003; Verburg et al., 2004; Gu et al., 2008). Second, the changed soil moisture scheme increased the moisture modifier ($\xi$w) on heterotrophic respiration during the peak-growing season, and decreased it during the non-peak growing season (Fig. S2), which is consistent with the trend of changes in SWP (Fig. 3). In addition, the changed soil moisture scheme also increased the simulated SOC stock, the substrate for heterotrophic respiration (Fig. S3). These changes together resulted in the improvement of simulated SR."*

[Figure]

**Figure S1: Relationship between changes in simulated annual soil respiration ($\Delta$SR) and gross primary production ($\Delta$GPP) induced by improvement of soil water potential using the Hanson model.**

[Figure]

**Figure S2 Impact of the changed SWP on the moisture modifiers of GPP (btran, a) and heterotrophic respiration ($\xi_w$, b).** $MOD_{default}$: model output before soil water potential improvement; $MOD_H$: model output after soil water potential improvement using the Hanson model.

[Figure]

**Figure S3 Comparison of the observed and modeled soil organic carbon (SOC) stocks.** OBS: observation; MOD: model output before soil water potential improvement; $MOD_H$: model output after soil water potential improvement using the Hanson model; $MOD_{H\_param}$: model output after soil water potential improvement using the Hanson model and parameter adjustments.

*Major concerns*
*I appreciate the effort used to explore alternative formulations for SWP in the model (Fig. 1, Table 1). Two questions come to mind. First, is it worth doing a more thorough model selection process like AIC or BIC that penalizes more complex models for their additional parameters instead of just showing RMSE. Second, as E3SM is intended to be run in global simulations, I wonder what effect alternative formulations for SWP have on water and energy fluxes from the model in site level, and ultimately global, simulations? The GPP results (Fig. 2) are a good start for this, but presumably these changes really modify ET fluxes (and runoff). It seems documenting these changes are likely important (if only in SI)?*

**Response:** The reviewer provided valuable comments in terms of the model selection and the effect of the alternative formulations for SWP on water and energy fluxes. In the revised manuscript, we added Akaike Information Criterion (AIC), in addition to root-mean-square-error (RMSE), for the model selection. Both AIC and RMSE indicated that the Hanson model was the best in simulating the SWP-VWC relationship (i.e., smallest AIC and RMSE values; revised Table 1 and also see below).

**Table 1.** Root-mean-square-error (RMSE) and Akaike Information Criterion (AIC) of different models in simulating the SWP-VWC relationship for the soil in the MOFLUX site at two depths: 0 to 30 cm and below 30 cm.

| Model | < 30 cm | | > 30 cm | |
|---|---|---|---|---|
| | RMSE | AIC | RMSE | AIC |
| Clapp & Hornberger (default ELMv0) | 4.25 | 157.82 | 1.33 | 18.51 |
| Brooks & Corey | 3.91 | 151.05 | 1.13 | 13.51 |
| Clapp & Hornberger (calibrated) | 0.53 | -61.03 | 0.51 | -23.43 |
| Fredlund & Xing | 0.51 | -63.15 | 2.43 | 47.13 |
| Hanson | 0.41 | -86.07 | 0.34 | -38.98 |
| van Genuchten | 0.50 | -65.53 | 0.36 | -36.61 |

In addition, we analyzed changes in simulated evapotranspiration (ET) and runoff as suggested. We plotted the mean annual cycle (± 1 sigma) of both ET and runoff (Fig. S6 and also see below). The change in soil moisture scheme using the Hanson model and parameter adjustments slightly increased ET and decreased runoff. Despite these slight changes, the model-simulated ET generally fell within the observed range, with or without changes in soil water scheme and parameters.

In the revised manuscript, we added a paragraph in the Results section to describe the changes in ET, runoff and other variables (page 10, line 11 – 16):

*"In addition, we analyzed changes in simulated evapotranspiration (ET), runoff, photosynthesis, net primary production, C allocations to fine roots, leaf and woody tissue in response to the changes in the soil water scheme and parameters (Fig. S6, S7). The change in soil moisture scheme and parameter adjustments slightly increased ET and decreased runoff.*

*Despite these slight changes, the model simulated ET generally fell within the observed range, with or without changes in soil water scheme and parameters (Fig. S6)."*

[Figure]

**Figure S6 Modeled evapotranspiration (ET) and runoff in response to the improved SWP and parameter adjustments.** OBS: observation; MOD$_{default}$: model output before soil water potential improvement; MOD$_H$: model output after soil water potential improvement using the Hanson model; MOD$_{H\_param}$: model output after soil water potential improvement using the Hanson model and parameter adjustments.

*If Hanson or van Genuchten formulations are 'better' fits to the observations, why aren't they used for GPP simulations in Fig. 2? What's the purpose of exploring alternative SWP schemes, if they don't follow through the C cycle simulations in the model? Reading the text on the bottom of page 7, however, maybe (MODswp) is using the Hanson scheme? If so, does the calibrated Clapp & Hornberger approach provide similar improvement by removing the high bias in the default configuration (Fig. 3). Please clarify in the text and figure captions what's being shown and why none of the models adequately capture the effect of the 2012 drought.*

**Response:** We apologize for the unclear description in the manuscript. The Hanson model was used through the carbon cycle simulations in the model. As shown in Fig. S2, the changed soil water scheme had impacts on both the moisture modifiers of GPP and heterotrophic respiration. We have revised the text and figure captions to make it clearer.

In response to the question *"does the calibrated Clapp & Hornberger approach provide similar improvement by removing the high bias in the default configuration"*, we conducted an additional analysis as described in the response to the first comment above. We compared the simulated gross primary production (GPP) and soil respiration (SR) when using the Hanson model and the calibrated Clapp & Hornberger model. Generally, both the Hanson model and the calibrated Clapp & Hornberger model improved the simulation of GPP and SR in the ELMv0, in comparison with the default run (Fig. S8 and also see below). The ELMv0 with the Hanson model consistently produced higher GPP and SR than that with the calibrated Clapp & Hornberger model. In comparison with the observations, the modeled SR generally fell within the 1 sigma range of observations, for both the Hanson model and the calibrated Clapp & Hornberger model. However, the modeled GPP with the calibrated Clapp & Hornberger model

was significantly lower than the observations. Given the goodness-of-fit of the soil water potential (SWP)-volumetric water content (VWC) relationship was default Clapp & Hornberger model < calibrated Clapp & Hornberger model < Hanson model (revised Table 1 and also see below), these new results further support our conclusion that better representations of SWP can improve the simulations of carbon processes (i.e., GPP and SR here).

In the revised manuscript, we added a new supplementary figure (i.e., Fig. S8) and a paragraph in the text to compare simulations with the Hanson model and the calibrated Clapp & Hornberger model (page 12, lines 10 – 19):

*"Moreover, we also explored whether the calibrated Clapp & Hornberger model can lead to similar improvements with the Hanson model (Fig. S8). Generally, both the Hanson model and the calibrated Clapp & Hornberger model improved the simulation of GPP and SR in the ELM, in comparison with the default run (Fig. S8). The ELMv0 with the Hanson model consistently produced higher GPP and SR than that with the calibrated Clapp & Hornberger model. In comparison with the observations, the modeled SR generally fell within the 1 sigma (i.e., standard deviation) range of observations, by using both the Hanson model and the calibrated Clapp & Hornberger model. However, the modeled GPP with the calibrated Clapp & Hornberger model was still lower than the observations. Given the goodness-of-fit of the soil water potential (SWP)-volumetric water content (VWC) relationship was default Clapp & Hornberger model < calibrated Clapp & Hornberger model < calibrated Hanson model (Table 1), these results further support the conclusion that better representations of SWP can improve the simulations of carbon processes. Therefore, throughout the remainder of this manuscript, we used the Hanson model to represent the SWP-VWC relationship."*

[Figure]

**Figure S8: Annual soil respiration (SR) and gross primary production (GPP).** Blue lines are the ELMv0 simulations with default parameters ($MOD_{default}$), red lines with the soil water potential improved using the calibrated Clapp & Hornberger model ($MOD_{cCP}$), and purple lines with the soil water potential improved using the Hanson model ($MOD_H$). Black lines and grey area are the observed (OBS) mean and 1 sigma range, which were calculated from 8 field replications for SR, and from three different net ecosystem exchange partitioning methods for GPP. The inserted bar plots are mean annual average ± 1 sigma across 2005-2011.

In response to the question "*why none of the models adequately capture the effect of the 2012 drought*", we revised Section 4.2 to include more discussion. Failing capturing the effect of the 2012 drought, as well as the underestimated seasonal and interannual variabilities of GPP and SR (Fig. 2, 4), indicate that the current model structure is not sensitive enough to environmental changes. Potential reasons include lacking representations of microbial organisms, macroinvertebrate and other forest floor and soil fauna and root exudates. We discussed in detail in Section 4.2 as:

[revised manuscript text omitted]

*Minor concerns*
*Section 2.3 This is really a broader comment on how author groups working with E3SM intend to articulate the version of the model on which they are working, esp. for readers less familiar with nuances of CLM4.5 development branches and subsequent ELM developments. For example, how is this code different from other publications (e.g. Brunke et al. 2016; Riley 2018)?*

*Section 2.3. Please justify the decision to use the CLM-CN decomposition module for a paper focused on soil respiration when Bonan and others (2013) clearly demonstrated shortcomings of this model version? It seems like the wrong tool for this job?*

*Section 2.3. This is also a little confusing, as the opening line of the section states the soil biogeochemistry is vertically resolved, but to my knowledge CLM-CN does not apply vertically resolved soil BGC? Please clarify*

**Response:** We appreciate the detailed comments on the model version. We used the ELM version 0 (ELMv0), which is equivalent to the Community Land Model version 4.5 (CLM 4.5). In ELMv0, the soil biogeochemistry can be simulated with one-layer or multi-layer converging trophic cascade (CTC, i.e., CLM-CN) decomposition model. We used the vertically-resolved CTC decomposition in this study. . Variable thickness of the soil profile in Brunke et al. (2016) and lateral energy and hydrological exchanges in Bisht et al. (2018) (which may be paper (Riley, 2018) the reviewer referred to) were not in the ELMv0.

In response to the comment "*It seems like the wrong tool for this job*", we used the ELMv0, which is structurally equivalent to the CLM 4.5. Recently, the E3SM council completed a comprehensive study of the soil biogeochemistry module by benchmarking different approaches with global (e.g., ILAMB) and Ameriflux datasets. Due to satisfactory overall performance of the CTC approach (i.e., CLM-CN), the council recommended the CTC approach as the default and baseline soil decomposition pathway in future ELM development. Therefore, we decided to use the CTC decomposition pathway in each soil layer for our study.

In the revised manuscript, we revised this section as (page 3, line 27 – 29):

*"The ELMv0 used in this study is structurally equivalent to the Community Land Model 4.5 (CLM 4.5), which includes coupled carbon and nitrogen cycles (Oleson et al., 2013). In ELMv0, the soil biogeochemistry can be simulated with one-layer or multi-layer converging trophic cascade (CTC, i.e., CLM-CN) decomposition model. We used the vertically-resolved CTC decomposition in this study."*

*Page 4, Line 20. Single point runs (especially with CLM) forced with flux tower measurements have a long history that should be acknowledged here.*

**Response:** We add a sentence to acknowledge the long history of single point runs (page 4, line 20 – 22):

*"Single-point runs forced with site-level measurements have a long history to evaluate model representations of phenology, net primary production (NPP), transpiration, leaf area index (LAI), water use efficiency, and nitrogen use efficiency (Richardson et al., 2012; De Kauwe et al., 2013; Walker et al., 2014; Zaehle et al., 2014; Mao et al., 2016; Duarte et al., 2017; Montané et al., 2017)."*

*Page 7, Line 7. What changes were made to the Clapp and Hornberger parameterization, there are lots of hard coded parameters in eq. 11-13.*

**Response:** For the calibration of the Clapp & Hornberger model, instead of using the hard-coded parameters in Eq. 11-13, we calibrated the three parameters (i.e., $\Psi_m$, $\theta_s$ and $\Psi_s$) in the Clapp & Hornberger model (Eq. 10). We added the sentence in the **Materials and Methods** section (page 6, line 5 – 7):

*"For the calibration of the Clapp & Hornberger model, instead of using the hard-coded (default) parameters in Eq. 11-13, we calibrated the three parameters (i.e., $\Psi_m$, $\theta_s$ and $\Psi_s$) in the Clapp & Hornberger model (Eq. 10)."*

*Fig 2. Why are observations shown with a black line and purple bar (inset)? Consistency within and among figures will help readers understand display items more easily? Similarly, using the same color for line of the default model and modified model in Figs. 1 and 2 would be helpful One strength of using flux tower data in single point simulations seems to be examining the seasonal cycle of carbon and energy fluxes. This is somewhat lost in Fig. 6, and I wonder if the display item would be more powerful if simulations are results were averaged over the whole*

*observation record (e.g. just show 1 year instead of 9, as the interannual variability isn't that obvious (and already shown in Fig. 2)*

**Response:** We really appreciate the great suggestion on the figure display. In the revised manuscript, we used consistent colors. In addition, we re-plotted Fig. 6 (revised Fig. 4) as the mean annual cycle (with 1 sigma) as suggested here and in a few other comments.

*Fig 4 is never really discussed and doesn't add much to the paper in my estimation. Can it be removed from the text? More, it follows that that changes in productivity would have a linear effect on soil C stocks and therefore respiration rates in a first order model like CLM-cn (Todd-Brown refs from the text), so the relationship shown here isn't really surprising.*

**Response:** We moved Fig. 4 to Fig. S1 as discussed above.

*Out of curiosity, how do simulated soil (or vegetation) C stocks compare with observed stocks at the site? The focus on fluxes is fine, but given that fluxes are linear related with stocks, do suggested modifications to the model improve estimates of fluxes AND stocks for the site?*

**Response:** This is really an insightful comment. We analyzed the modeled soil organic carbon (SOC) stocks in the revised manuscript (Fig. S3 and also see below). Although the improved SWP simulations increased SOC stocks, the model simulated SOC with different soil water schemes and parameters generally fell within the wide range of observations.

[Figure]

**Figure S3 Comparison of the observed and modeled soil organic carbon (SOC) stocks.** OBS: observation; MOD: model output before soil water potential improvement; $MOD_H$: model output after soil water potential improvement using the calibrated Hansom model; $MOD_{H\_param}$: model output after soil water potential improvement using the calibrated Hansom model and parameter adjustments.

In the revision, we added the description of the result as:

*"While the model simulated SOC with different soil water schemes generally fell within the wide range of observations, the improved SWP simulations increased SOC stocks (Fig. S3)."*

*Seasonal biases in SR and GPP fluxes look pretty bad with default and 'swp' versions of the model (Fig. 5). The parametric changes in Table 2 seem to address some of these seasonal biases (Fig. 6), but it seems like showing the scatter plots on Fig. 5 (maybe with a 3rd color) would be helpful? Along these lines, should both Figs. 5 and 6 show the same 3 simulation ('default', 'swp', and 'swp_param')? Showing the mean annual cycle (+/- 1 sigma on the observations) for all panels in Fig. 6 would help to make this figure easier to digest.*

**Response:** In the revision, we re-plotted Fig. 6 (revised Fig. 4 and also see below) as the mean annual cycle (with 1 sigma) as suggested. We included all 'default', 'H', 'H_param', and 'obs' in the figure. The revised figure can clearly show the seasonal biases in SR and GPP fluxes. Thus, the original Fig. 5 was duplicated by this presentation, so we deleted Fig. 5 in the revised manuscript. Because we moved the original Fig. 4 to Fig. S1, the original Fig. 6 is Fig. 4 in the revised manuscript.

[Figure]

**Figure 4 The annual mean cycles of leaf area index (LAI), gross primary production (GPP) and soil respiration (SR).** OBS: observation; MOD$_{default}$: model output before soil water potential improvement; MOD$_H$: model output after soil water potential improvement by the Hanson model; MOD$_{H\_param}$: model output after soil water potential improvement by the Hanson model and parameter adjustments.

*SLA is something that's measured, maybe not at the site for similar trees to the ones at the site? Is building 3x thicker leaves (Table 2), a reasonable assumption? Similarly, if the authors need to decrease LAI while increasing GPP, flnr necessarily has to increase in the model, but is the 20% increase here supported by databases like TRY, or are these parameter changes just illustrating big nobs in the model that are poorly constrained by observaions?*

**Response:** We appreciate the detailed suggestion on parameter values. Unfortunately, SLA was not measured at the site. The parameter adjustments were based on a surrogate based global optimization using measurements of C and energy fluxes at the site (Lu et al., 2018). The TRY database showed that the SLA for broadleaved deciduous forest ranges from < 0.0005 to > 0.005

m$^2$ g$^{-1}$ C, with mean values of 0.015 m$^2$ g$^{-1}$ C (Kattge et al., 2011). Thus, the adjustment of the parameter *slatop* fell within the range of observations.

*Page 10, line 12 please report statistics to support claims being made. Visually, the red line looks closer to the observations than the blue one (Fig. 6b,c). How do the annual totals look?*

*As with comment above, how do changes in annual fluxes or total stocks compare with observations following parametric changes suggested in Table 2?*

**Response:** We replotted Fig. 6 (Fig. 4 in the revision) to show the annual cycle ± 1 sigma considering all 9 years of data. With the revised figure, we can methodically step through the changes. Results showed that both the default and improved SWP using the Hanson model overestimated the maximum LAI (Fig. 4a). The parameter changes significantly reduced the maximum LAI to better match the observations (Fig. 4a). The parameter changes further increased the simulated GPP and SR during the peak growing season, in addition to the improvement by the adjusted SWP (Fig. 4b, c). However, the ELMv0 still overestimated SR during the non-peak growing season.

[Figure]

**Figure 4 The annual mean cycles of leaf area index (LAI), gross primary production (GPP) and soil respiration (SR).** OBS: observation; MOD$_{default}$: model output before soil water potential improvement; MOD$_H$: model output after soil water potential improvement by the Hanson model; MOD$_{H\_param}$: model output after soil water potential improvement by the Hanson model and parameter adjustments.

To answer the question "*how do the annual totals look*", we analyzed the mean annual fluxes of GPP and SR (Fig. S5 and also see below) and SOC stocks (Fig. S3 and also see below). After the parameter adjustments, GPP was still within the observed ranges, while SR was significantly overestimated due to the overestimation of SR during the non-peak growing season. In Section 4.2, we discussed the potential reasons, including Q$_{10}$ and representations of microbial organisms, macroinvertebrate and other forest floor and soil fauna.

[Figure]

**Figure S5 Comparison of the observed and modeled gross primary production (GPP) and soil respiration (SR).** OBS: observation; MOD: model output before soil water potential improvement; $MOD_H$: model output after soil water potential improvement using the calibrated Hansom model; $MOD_{H\_param}$: model output after soil water potential improvement using the calibrated Hansom model and parameter adjustments.

[Figure]

**Figure S3 Comparison of the observed and modeled soil organic carbon (SOC) stocks.** OBS: observation; MOD: model output before soil water potential improvement; $MOD_H$: model output after soil water potential improvement using the calibrated Hansom model; $MOD_{H\_param}$: model output after soil water potential improvement using the calibrated Hansom model and parameter adjustments.

In the revision, we added the new results (page 10, line 6 – 10):

[revised manuscript text omitted]

*Page 12, line 4. This doesn't seem like a fair statement or comparison, as results from the tuned Clapp & Hornberger scheme are never presented.*

**Response:** In the revision, we compared the ELMv0 simulations with the Hanson model and the calibrated Clapp & Hornberger scheme as described above. The performance of the Hanson model was better than the calibrated Clapp & Hornberger model. Thus, we used the Hanson model to improve the SWP simulations.

*Page 12, line 10, given the dominance of Rh in contributions to soil respiration (Fig 7). I'd suspect that changes in SR have more to do with larger SOM stocks than they do links between substrate supply through GPP, as suggested here, but no data are presented along these lines?*

**Response:** As described above, the improved SR was a joint result of changes in GPP, SOC stocks and the moisture modifier of heterotrophic respiration. We revised this part (page 11, line 1 – 11) as:

*"The improvements in model fits were due to changes in GPP, SOC stocks, and the moisture modifier on heterotrophic respiration (Figs. S1 – S3). First, the default ELMv0 underestimated GPP (Fig. 2b), as in a recent study where CLM4.5 significantly underestimated GPP at a coniferous forest in northeastern United States (Duarte et al., 2017). GPP affects the substrate supply for SR, as evidenced by the close relationship between changes in SR and GPP (Fig. S1),*

*which is consistent with experimental evidence showing GPP can directly affect the magnitude of root respiration (Craine et al., 1999; Högberg et al., 2001; Wan and Luo, 2003; Verburg et al., 2004; Gu et al., 2008). Second, the changed soil moisture scheme increased the moisture modifier (ξw) on heterotrophic respiration during the peak-growing season, and decreased it during the non-peak growing season (Fig. S2), which is consistent with the trend of changes in SWP (Fig. 3). In addition, the changed soil moisture scheme also increased the simulated SOC stock, the substrate for heterotrophic respiration (Fig. S3). These changes together resulted in the improvement of simulated SR."*

*Page 12, line 20. This statement may be true, but it's not clear that changes to VMC proposed here had much of an effect on the Rh component of the model. To show this, it seems like showing the soil moisture effect (w_scalar) on soil decomposition rates from different configurations of the model would be needed. Otherwise, I'd suspect that improvement to SR (Fig 2, 5) are predominantly driven by larger soil C stocks (via higher GPP), but not from direct improvement in the SWC on soil biogeochemistry, as suggested in the Powell paper referenced.*

**Response:** The soil moisture modifier (w_scalar, which is $\xi w$ in Eq. 9) of heterotrophic respiration was shown in Fig. S2 (and also see below) in the revision. In addition, the SOC stocks under different schemes were shown in Fig. S3. As discussed above, the improvement of SR was a joint result of changes in GPP, SOC stocks and the moisture modifier.

[Figure]

**Figure S2 Impact of the changed SWP on the moisture modifiers of GPP (btran, a) and heterotrophic respiration ($\xi$w, b).** $MOD_{default}$: model output before soil water potential improvement; $MOD_H$: model output after soil water potential improvement using the Hanson model.

*Page 12, line 20 what is SWP-VMC, should be VWC?*

Response: Yes and revised.

*Page 13, line 2. Again this claim is poorly supported by the data presented. Yes, the tuned Clapp and Hornberger model is not the 'best' model in Table 1, but are results for GPP or SR markedly different than the Hansen results shown?*

**Response:** Yes, the results for GPP and SR were markedly different when using the Hanson model and the calibrated Clapp & Hornberger model as discussed above. The ELMv0 with the Hanson model consistently produced higher GPP and SR than that with the calibrated Clapp & Hornberger model. In comparison with the observations, the modeled SR generally fell within the 1 sigma (i.e., standard deviation) range of observations, by using both the Hanson model and the calibrated Clapp & Hornberger model. However, the modeled GPP with the calibrated Clapp & Hornberger model was lower than the observations.

In response to the reviewer's concern, we deleted the sentence in the revision.

*Page 14. The q10 analysis is nice, but I wonder if a more ecological explanation is relevant here- specifically highlighting the role of root exudates in supplying labile C substrates that are important for SR? The land model here doesn't consider these ecologically important C fluxes that likely have an important control over the seasonal dynamics of soil respiration and microbial biomass already discussed?*

Response: The reviewer provided a great potential reasons for the modeled biases of SR. In the revision, we added the discussion of root exudates as:

*"Our analyses also showed that the modeled SR was not able to reach the observed peak in many years during the peak growing season even when the modeled GPP exceeded the observation (Fig. S11). In addition, the parameter modification increased GPP during both peak and non-peak growing seasons, resulting in an even greater overestimation of SR during non-peak growing seasons (Fig. S11). These results suggest that simply increasing GPP may not be adequate to increase the seasonal variability of the simulated SR. A potential reason may be that the current model does not include root exudates. Root exudates are labile C substrates that are important for SR (Kelting et al., 1998; Kuzyakov, 2002; Sun et al., 2017). The root exudate rate is primarily dependent on root growth, showing a seasonal cycle in temperate forests (Kelting et al., 1998; Kuzyakov, 2002). Thus, including root exudates in the model may further increase the model simulated SR during the peak growing season without needing to increase GPP."*

*References: Bonan, et al (2013) Evaluating litter decomposition in earth system models with long-term litterbag experiments: an example using the Community Land Model version 4 (CLM4), Global Change Biology, 19, 957-974.*
*Brunke et al. 2016. "Implementing and Evaluating Variable Soil Thickness in the Community Land Model, Version 4.5 (CLM4.5)." Journal of Climate 29(9): 3441-3461, doi:10.1175/JCLI-D-15-0307.1.*

*Riley, W. 2018. "Impacts of Microtopographic Snow Redistribution and Lateral Sub- surface Processes on Hydrologic and Thermal States in an Arctic Polygonal Ground Ecosystem: A Case Study Using ELM-3D v1.0." Geoscientific Model Development 11(1): 61-76, doi:10.5194/gmd-11-61-2018.*

**Response:** We appreciate the information!

*Reviewer 2*

*This paper reports an effort of tuning an Earth system model, E3SM, to fit observed leaf area index (LAI), gross primary production (GPP, derived from eddy flux data), and soil respiration at a temperate deciduous forest site. The authors specifically tested different empirical relationships between volumetric water content (VWC) and soil water potential (SWP), and found tuning soil water potential improve the simulation of soil respiration. So, they concluded that "modelling soil respiration can be significantly improved by better model representations of the soil water retention curve." I agree with the authors that the well data-constrained model, Hanson model, increased the prediction of soil water potential, and may improve the simulation of GPP, which have been shown by the results (Figs. 3 and 7). But for the improvement of soil respiration, I think it's just a coincidence. From the Fig. 5a (page 9), we can see the new VMC-SWP relationship (i.e., Hanson model) increases soil respiration rate overall, but it does NOT change the pattern. This means the performance of soil respiration modeling is not improved. The authors also pointed out that the original model underestimates GPP and soil respiration (Line 13, page 7, and Fig. 2). So, the improvement of soil respiration prediction was not due to the improvement of SWP simulation, but because increases in GPP. The increases in GPP may increase carbon allocation to roots or total soil carbon, and therefore increase soil respiration. And, according to Fig. 7, the most possible reason for underestimating soil respiration is that the root respiration is not high enough in growing season, which also leads to the seasonal pattern that does not fit the observations because root respiration is usually high in growing season and very low in non-growing season.*

**Response:** We greatly appreciate the valuable comment. The reviewer criticized that "*the improvement of soil respiration prediction was not due to the improvement of SWP simulation, but because increases in GPP*". A similar comment was raised by the first reviewer who commented that "*it's not clear if this is a direct effect of soil moisture on soil respiration*". We appreciate both the reviewers pointed the issue out. In response to the comments, we analyzed the changes in the soil moisture modifiers of GPP (btran) and heterotrophic respiration ($\xi_w$) and soil organic carbon (SOC) (Fig. S2-S3 and also see below), as suggested by the first reviewer. The improved soil water scheme using the Hanson model increased both *btran* and $\xi_w$ during the peak growing season, and reduced $\xi_w$ during the non-peak growing season (Fig. S2). While the model-simulated SOC with different soil water schemes generally fell within the wide range of observations, the improved SWP simulations increased SOC stocks (Fig. S3). These results, combining with previous results (revised Fig. S1), indicate that the improved annual fluxes of SR by SWP was a joint result of changes in GPP, SOC stocks and the moisture modifier on heterotrophic respiration.

In the revised manuscript, we added two new supplementary figures (Fig. S2, S3), moved Fig. 4 as Fig. S1, presented the results (page 8, line 1 – 5) and discussed details (page 11, line 1 – 11) in the text.

*"The changes in annual SR and GPP (i.e., the differences between before and after the improved SWP simulation using the Hanson model) showed a linear relationship (Fig. S1). In addition, the improved soil water scheme using the Hanson model increased both the moisture modifiers of GPP and heterotrophic respiration (i.e., btran and $\xi_w$) during the peak growing season, and reduced $\xi_w$ during the non-peak growing season (Fig. S2). While SOC when simulated by the model with different soil water schemes generally fell within the wide range of observations, the improved SWP simulations using the Hanson model increased SOC stocks (Fig. S3)."*

*"Constraining the SWP-VWC relationship with site-specific data and using the Hanson model instead of the ELMv0 default model (Fig. 1) significantly improved the model representation of SWP (Fig. 3) and annual SR (Fig. 2a). The improvements in model fits were due to changes in GPP, SOC stocks, and the moisture modifier on heterotrophic respiration (Figs. S1 – S3). First, the default ELMv0 underestimated GPP (Fig. 2b), as in a recent study where CLM4.5 significantly underestimated GPP at a coniferous forest in northeastern United States (Duarte et al., 2017). GPP affects the substrate supply for SR, as evidenced by the close relationship between changes in SR and GPP (Fig. S1), which is consistent with experimental evidence showing GPP can directly affect the magnitude of root respiration (Craine et al., 1999; Högberg et al., 2001; Wan and Luo, 2003; Verburg et al., 2004; Gu et al., 2008). Second, the changed soil moisture scheme increased the moisture modifier ($\xi_w$) on heterotrophic respiration during the peak-growing season, and decreased it during the non-peak growing season (Fig. S2), which is consistent with the trend of changes in SWP (Fig. 3). In addition, the changed soil moisture scheme also increased the simulated SOC stock, the substrate for heterotrophic respiration (Fig. S3). These changes together resulted in the improvement of simulated SR."*

[Figure]

**Figure S1: Relationship between changes in simulated annual soil respiration (ΔSR) and gross primary production (ΔGPP) induced by improvement of soil water potential using the Hanson model.**

[Figure]

**Figure S2 Impact of the changed SWP on the moisture modifiers of GPP (btran, a) and heterotrophic respiration ($\xi_W$, b).** $MOD_{default}$: model output before soil water potential improvement; $MOD_H$: model output after soil water potential improvement using the Hanson model.

[Figure]

**Figure S3 Comparison of the observed and modeled soil organic carbon (SOC) stocks.** OBS: observation; MOD: model output before soil water potential improvement; $MOD_H$: model output after soil water potential improvement using the Hanson model; $MOD_{H\_param}$: model output after soil water potential improvement using the Hanson model and parameter adjustments.

*A detailed report on the tuning of an ESM is valuable even if no new mechanisms were added. It helps to understand model performance and the thoughts behind the model development. For improving simulation of soil respiration, the authors had looked at the sensitivity to temperature, LAI, GPP, and relative contributions of roots and soil carbon, and tuned a bunch of parameters (Table 2 in page 5). A detailed analysis of the successes and fails of these tunings would be interesting. For example, I'd like to see how the improvement of SWP prediction affects plant physiology, photosynthesis, allocation, NPP (because NPP=Rh at equilibrium). These variables may change soil respiration.*

**Response:** The reviewer provided an insightful suggestion. In the revised manuscript, we added the analyses of the effects of the improved SWP on photosynthesis, NPP, and carbon allocation to fine root, leaf and woody tissue (Fig. S7 and also see below). Results showed that the improved SWP generally increased all photosynthesis, NPP and carbon allocations to different tissues during the growing season. In addition, parameter adjustments further increased them.

[Figure]

**Figure S7 The annual mean cycles of photosynthesis (Pn), net primary production (NPP) and C allocations to fine root (Allocation_froot), leaf (Allocation_leaf) and woody tissue (Allocation_wood).** MOD_default: model output before soil water potential improvement; MOD_H: model output after soil water potential improvement using the Hanson model; MOD_H_param: model output after soil water potential improvement using the Hanson model and parameter adjustments.

We added the description of these results in the revised manuscript as (page 10, line 11 − 16):

*"In addition, we analyzed changes in simulated evapotranspiration (ET), runoff, photosynthesis, net primary production, and C allocation to fine roots, leaf and wood in response to the changes*

*in the soil water scheme and parameters (Fig. S6, S7). The change in soil moisture scheme and parameter adjustments slightly increased ET and decreased runoff. Despite these slight changes, the model simulated ET generally fell within the observed range, with or without changes in soil water scheme and parameters (Fig. S6). The improved SWP and parameter adjustments generally increased all photosynthesis, NPP and carbon allocations to different tissues during the growing season (Fig. S7)."*

*Specifically, for water effects on soil heterotrophic respiration, the model uses two equations to link volumetric water content to heterotrophic respiration: VMC−>SWP and SWPRh. The second equation (SWPRh, Eq 9 in page 4) is much more critical than the first one for modeling heterotrophic respiration. It represents the knowledge of how soil moisture affects microbial physiology. It needs to be explored in detail if the goal of this research is to improve the simulation of soil respiration.*

**Response:** We agree with the reviewer that the moisture modifier ($\xi$w) is a critical factor in the model in determining how soil moisture affects microbial physiology. In the revised manuscript, we analyzed the effect of the improved SWP on $\xi$w (Fig. S2b). The improved SWP increased $\xi$w during the peak growing season, and reduced $\xi$w during the non-peak growing season (Fig. S2), which was consistent with the changes in SWP (Fig. 3b). The changes in $\xi$w, as well as GPP and SOC stocks, jointly determined the effect of the improved SWP on SR, as discussed above and in the revised manuscript.

[Figure]

**Figure S2 Impact of the changed SWP on the moisture modifiers of GPP (btran, a) and heterotrophic respiration ($\xi$w, b).** MOD$_{default}$: model output before soil water potential improvement; MOD$_H$: model output after soil water potential improvement using the Hanson model.

The related revisions are shown below and in the text (page 8, line 1 – 5; page 11, line 1 – 11).

*"The changes in annual SR and GPP (i.e., the differences between before and after the improved SWP simulation using the Hanson model) showed a linear relationship (Fig. S1). In addition, the improved soil water scheme using the Hanson model increased both the moisture modifiers of GPP and heterotrophic respiration (i.e., btran and $\xi_w$) during the peak growing season, and reduced $\xi_w$ during the non-peak growing season (Fig. S2). While SOC when simulated by the model with different soil water schemes generally fell within the wide range of observations, the improved SWP simulations using the Hanson model increased SOC stocks (Fig. S3)."*

*"Constraining the SWP-VWC relationship with site-specific data and using the Hanson model instead of the ELMv0 default model (Fig. 1) significantly improved the model representation of SWP (Fig. 3) and annual SR (Fig. 2a). The improvements in model fits were due to changes in GPP, SOC stocks, and the moisture modifier on heterotrophic respiration (Figs. S1 – S3). First, the default ELMv0 underestimated GPP (Fig. 2b), as in a recent study where CLM4.5 significantly underestimated GPP at a coniferous forest in northeastern United States (Duarte et al., 2017). GPP affects the substrate supply for SR, as evidenced by the close relationship between changes in SR and GPP (Fig. S1), which is consistent with experimental evidence showing GPP can directly affect the magnitude of root respiration (Craine et al., 1999; Högberg et al., 2001; Wan and Luo, 2003; Verburg et al., 2004; Gu et al., 2008). Second, the changed soil moisture scheme increased the moisture modifier ($\xi_w$) on heterotrophic respiration during the peak-growing season, and decreased it during the non-peak growing season (Fig. S2), which is consistent with the trend of changes in SWP (Fig. 3). In addition, the changed soil moisture scheme also increased the simulated SOC stock, the substrate for heterotrophic respiration (Fig. S3). These changes together resulted in the improvement of simulated SR."*

*Short comment 1:*

*Dear authors,*
*in my role as Executive editor of GMD, I would like to bring to your attention our Editorial version 1.1:*

*http://www.geosci-model-dev.net/8/3487/2015/gmd-8-3487-2015.html*

*This highlights some requirements of papers published in GMD, which is also available on the GMD website in the 'Manuscript Types' section:*

*http://www.geoscientific-model-development.net/submission/manuscript_types.html*

*In particular, please note that for your paper, the following requirement has not been met in the Discussions paper:*

*• "The main paper must give the model name and version number (or other unique identifier) in the title."*

*Even if this is not a strict requirement for evaluation paper, we like to encourage authors to provide also the version number of the evaluated model, as usually evaluation results depend on model version.*

*Additionally, please not, that GMD is encouraging authors to provide a persistent access to the exact version of the source code used for the model version presented in the paper. As explained in https://www.geoscientific-modeldevelopment.net/about/manuscript_types.html the preferred reference to this release is through the use of a DOI which then can be cited in the paper. For projects in GitHub (such as thee E3SM Land Model) a DOI for a released code version can easily be created using Zenodo, see https://guides.github.com/activities/citable-code/ for details.*

*Yours,*
*Astrid Kerkweg*

**Response:** We appreciate the comments. We revised the title to include the model name and version number. The revised title is *"Evaluating the E3SM Land Model version 0 (ELMv0) at a temperate forest site using flux and soil water measurements"*. We also provided the link to the model.

[revised manuscript text omitted]

---

## Referee Report (RR1)

Review: Evaluating the E3SM Land Model Version 0 (ELMv0) at a temperature forest site using flux and soil water measurements

Authors: Liang, Wang, Ricciuto, Gu, Hanson, Wood, Mayes

**Synopsis:** The authors run an out-of-the-box simulation of E3SM (which is the same as CLM4.5-CN; I'm not sure what makes E3SM distinct) and compare results to Eddy Covariance (EC) fluxes and soil moisture observations. The default model is found to have Gross Primary Productivity (GPP) and Soil Respiration (SR) that are too low when compared to observations. Near-surface Soil Water Potential (SWP), calculated using relationships that determine SWP from Volumetric Water Content (VWC) based on Clapp&Hornberger result in potentials that are too high during the winter, and too low in the summer. Overall, SWP is too low at low moisture in the near-surface soil, and was slightly high at depth when moisture content was higher.

Five different treatments for relating VWC-SWP were tested, and the model of Hanson was found to have the smallest errors when compared to observations. However, when C&H was replaced with Hanson GPP was slightly low and SR slightly high when compared to observations, and the model did not reproduce either the amplitude or sign of interannual variability. Therefore, coefficients influencing Specific Leaf Area (SLA), fractional leaf N used in Rubisco and several coefficients controlling leaf senescence were changed, and results were improved in evaluation of mean seasonal cycles of LAI, GPP and SR.

Finally, there was speculation about which mechanisms and processes might be responsible for model-data mismatch after the aforementioned tuning was complete. These include model Q10 for heterotrophic respiration, microbial biomass and seasonality, and macroinvertebrate (earthworm) influence on carbon cycle processes, and root exudates. The authors exhort the community to pay particular attention to SWP in simulations, and to consider inclusion of these added processes in models.

**Review:** One could consider this a model-tuning paper. A default model was run, deficiencies were noted, and changes were made to parameters and model physics. This is fine, and has been done many times previously (e.g. Sellers et al. 1989), but I'm not sure that the present paper really tells me anything about how the world works. I work with models that simulate land-atmosphere interaction, and there is nothing in this paper that makes me want to look at my model code and start performing tests and making changes. Hanson worked better, but it worked better at *one place on the planet,* at a particular deciduous forest (DBF) in the North American midwest. I suspect that if we were to perform evaluations like this at multiple EC sites (across multiple DBF sites and across multiple PFTs), I expect that we would find that each of the VWC-SWP treatments would come out on top at least one or more times. We would also likely find that the SLA, Nitrogen and senescence parameters could take multiple values as well.

My main complaint now levied, I will also say that just because this paper does not particularly *excite* me, there is nothing *wrong* with the analysis. The paper follows a logical progression, and the analysis and presentation of results is done professionally and is easy to follow. I think there is value in the paper, and my official recommendation is to accept this manuscript with minor revisions.

The paper is quite short (14 pages), which is nice, but I think there might be some expansion of analysis and explanation that would add value to the research.

Merely stating that the model was unable to capture observed response to the 2012 drought is extremely unsatisfying. This is an *opportunity* to explore model behavior, and perhaps gain valuable insight into processes and mechanisms. I find it interesting that observed near-surface (10cm) SWP (Fig 3b) was not exceptionally low in 2012; the year did not look much different from 2011 or 2013, and in fact looked wetter than 2005-2007. That is interesting; what was deep soil SWP doing in those years? From Figure 2, we see that observed SR oscillated up and down between 2005-2007, while GPP dropped from 2005 to 2007. What was the model doing? What did BTRAN look like in 2012, as compared to other years? How about LAI? Is there a near-surface water table in the simulations that prevents root stress? Are there constraints on stomatal conductance due to high VPD or unfavorable temperature? What did they do in 2012? In fact, simulated SR and GPP both increased from 2011 to 2012, while there were dramatic drops in the observations of both. If the model does not respond to the drought, you should be able to tell your reader *why*, and speculate whether that behavior is realistic or not, and how that behavior might impact model performance in other years. I would like to see some exploration of IAV, and explanation of why Hanson provides an upgrade from C&H in this regard.

Related to the above is the fact that in 2007 there was a significant drop in observed SR and GPP when compared to 2006. The default model (C&H) showed drops that are more similar to the amplitude of the observed reduction, even if there is an offset or bias. In fact, the Hanson model shows almost no interannual variability (IAV) in SR and GPP at all. Is this really an improvement? One might make the case that you would have a better simulation of the observed flux by increasing Vcmax (and perhaps SLA and the senescence parameters) in CLM without changing the soil; your GPP would go up, which would translate into larger carbon pools and subsequent increased RS. You would also retain a more realistic comparison with observed IAV. Is there a reason to suspect that this would not work?

In section 4.1 two paragraphs (lines 30-33 on page 11, continued in lines 1-7 on page 12; lines 8-13 on page 12) describe how C&H was developed from textural classes and not sand/clay fractions, and how models might make use of near-continuous SWP observations. I'm not sure what these 2 paragraphs bring to the analysis, since neither has been done. Do they merit this much attention?

The authors state that "SWP in simulations in ESMs should be calibrated carefully with observations…", but this is clearly impossible and unrealistic in global simulations. If the

ultimate goal is correct simulations of global biogeophysical behavior, then we have a disconnect between what the authors are doing here (tuning at a single site) and what we are told is the ultimate goal (accurate representation of global carbon cycle). This is a persistent and real problem. We calibrate our models on site-level data and then extend that behavior to the globe. I'd be interested in some discussion of how we might use site-level studies to improve global simulations.

**Specific Comments:**

- Equation 4: The subscript should be liveCroot, should it not?
- Page 4, line 9: coarse
- Page 5: is 'residual' water content the same as wilt point?
- Table 1: I'm not sure what AIC is: shouldn't it be explained, even if briefly?
- Page 8, line3: I know what 'btran' is, but some of your readers may not. You should explain this variable.
- Figure 4, Figures S6-S7: Make lines darker, shading lighter. Hard to discern individual simulations.
- I might have missed this, but what is the porosity and sand/clay content of the soil at the MOFLUX site? If VWC at depth regularly drops to between 15-20% (Fig 1b) then it must have considerable sand content. My recollection of more clayey soils is that wilt point will be much higher. Is this soil representative of the region and/or PFT?
- Equation 9: It appears that the environmental modifier for water has value of 0 and low water (conditions too dry for microbial activity), varies between 0-1 for moisture up to (PSI)max. What is the difference between (PSI)max and (PSI)s? Are they the same? Most models I am familiar with will have an 'optimum' soil water content or potential for respiration, the idea being that either too dry or too wet (anaerobic) conditions are unfavorable for microbial decomposition of carbon stocks. The 'too wet' does not seem to be the case here. Why is that?
- Using 10 years of tower forcing to perform a 200-year spinup of carbon pools concerns me. I understand that this might be all the tower data available, but, especially for carbon pools, I'm concerned that anomalies in the 10-year meteorology may be aliased onto pool size. Did you consider using a reanalysis product (CRU, NCEP, ECMWF) for spinup and then use the tower data for the transient run?

**References:**

Sellers, P.J., W.J. Shuttleworth, J.L. Dorman, A. Dalcher, J.M. Roberts, 1989: Calibrating the Simple Biosphere Model for Amazonian Tropical For- est Using Field and Remote Sensing Data. Part I: Average Calibration with Field Data. Journal of Applied Meteorology, 28, 727-759, August

1989.

---

## Referee Report (RR2)

Review: Liang et al., MOFLUX site, soil respiration

In general, the authors have addressed my main points from the initial review. This paper can be published, with the minor comments from below addressed. I do not need to see this paper again.

Specific comments:

Page 6, last line: It looks to me like C+H overestimates SMP when moisture is **below** 15% for depth below 30cm.

Page 10, line 4: There is no range on the GPP plot of Figure S5. Also, it looks like SR and GPP plots are reversed (a and b) when compared to their referencing in the text.

Page 12, line 5: "First, the Hanson Model significantly increased GPP." This is an incorrect statement. The Hanson model does not calculate GPP. The Hanson model determines SMP, and SMP is related to GPP through the ability of the model to capture soil moisture for use in transpiration. Interestingly, The Hanson SMP is only higher than the C+H SMP in the top 30cm of soil when VWC is < 15%. Everywhere else (top 30 cm, VWC > 15%, all VWC below 30cm) the Hanson SMP value is below (larger negative) than the C+H (Figure 1). What this says to me is that ELM transpiration (and GPP) is critically dependent on soil moisture in the upper 30 cm of the soil, and this upper soil is frequently at VWC below 15%. There should be some discussion of this. Do you think this result is realistic? What this is saying is that trees in the American Midwest are not at all dependent upon soil moisture below 30cm in the soil. Do you believe that to be true? Myself, I find this result suspicious. I was under the impression that deeper roots are critical to tree survival. This result contradicts that, as Hanson SMP was lower than C+H at all VWC below 30cm.

Section 4.2: The lack of IAV in the model may be tied to the dependence on near-surface soil moisture. It appears that the main change imposed by using the Hanson model is that SMP is higher, so moisture is more readily available to roots, in the upper 30cm of the soil, and only when VWC is below 15%. I would expect that VWC in the upper 30cm gets reduced to very small amounts every year. In that case the model would not be expected to see much IAV, ever. So it may be that the lack of IAV in the model has nothing to do with mortality rates or pathogens, and everything to do with how the model extracts water from the soil. On the other hand, Figure S10 shows that SWP was lowest during the entire record in the period 2005-2007, and the GPP in those years was not as extremely depleted as it was in 2012 (Figure 2). This is something the authors should discuss.

---

## Author Response (AR2)

**Letter of Response**

Dear Dr. Tomomichi Kato

We are re-submitting the manuscript entitled "Evaluating the E3SM Land Model at a temperate forest site using flux and soil water measurements" to be considered for publication in *Geoscientific Model Development*. We greatly appreciate the constructive comments from the two knowledgeable reviewers. We are also grateful to you for offering the opportunity to us to re-submit the manuscript, which has been thoroughly revised based on the reviewers' comments. We have made the following major changes:

1. We have conducted more analyses on the interannual variability and added more discussion in Section 4.2.
2. We have discussed in further detail on the future directions for the model improvement of water limitation to biogeochemical processes.
3. We have improved the presentations of figures and results by following the reviewers' suggestions.

We are submitting a letter of point-by-point responses to reviewers' comments, as well as a marked-up revision. We hope you will find our revision thorough and satisfactory.

Sincerely,
Junyi Liang

Author Note: The reviewers' comments are in black and the responses follow in blue. Page numbers and line numbers in the responses are those in the marked-up revision following the response letter.

Reviewer 1 (Dr. Will Wieder)

Liang and co-authors have done a nice job revising their manuscript. At its core, this is a paper about soil moisture, productivity and respiration. This is a hard problem, and one which could be handled more thoughtfully in the discussion.

Response: We appreciate the positive comments.

While I appreciate the qualifiers to text explaining to how "improving SWP directly improved soil respiration estimates" (top of page 8 & section 4.1), I would contend that the direct effects of SWP on respiration rates are overstated or unclear. It seems the improvements in soil respiration rates are driven by change in productivity (Fig. S1) which builds larger SOM pool. Specifically, it looks like at annual (Fig 2a, e.g. 2012) and seasonal (4c, shoulder seasons) scales the model shows a relatively low sensitivity to the modeled soil moisture scalar ($xi\_w$). I think this is OK, E3SM is in good company (see Carvalhais et al 2014)!

Response: We agree that the change in productivity is an important driving factor for the improvements in soil respiration, which is highlighted in the manuscript. As shown in the manuscript, improving SWP resulted in better GPP simulations, which improved soil respiration simulations in two aspects. On the one hand, GPP can directly affect the magnitude of root respiration. On the other hand, increased GPP can build larger SOM pools as the reviewer mentioned and shown in Fig. S3. In addition to GPP, we would argue that the change in SWP also has a direct impact on soil respiration as the soil moisture scalar ($\xi w$) controls the magnitude of heterotrophic respiration.

We revised the manuscript to incorporate the reviewer's comment as (Page 12, line 3 – 12):

> *"Constraining the SWP-VWC relationship with site-specific data and using the Hanson model instead of the ELMv0 default model (Fig. 1) significantly improved the model representation of SWP (Fig. 3) and annual SR (Fig. 2a). The improvements in model fits could be due to the following reasons. First, the Hanson model significantly increased GPP. The default ELMv0 underestimated GPP (Fig. 2b), as in a recent study where CLM4.5 significantly underestimated GPP at a coniferous forest in northeastern United States (Duarte et al., 2017). GPP can directly affect the magnitude of root respiration as shown in many previous studies (Craine et al., 1999; Högberg et al., 2001; Wan and Luo, 2003; Verburg et al., 2004; Gu et al., 2008). Additionally, increased GPP can build a larger SOC pool, the substrate for heterotrophic respiration (Fig. S3). Second, the changed soil moisture scheme increased the moisture modifier ($\xi_W$) on heterotrophic respiration during the peak-growing season, and decreased it during the non-growing season (Fig. S2), which is consistent with the trend of changes in SWP (Fig. 3). These changes together resulted in the improvement of simulated SR."*

The topic of soil moisture sensitivities is introduced again at the bottom of page 12 (section 4.2), but the discussion quickly moves onto temperature sensitivities and q10 before drifting off into soil microbes and fauna. As stated above, this is a paper about soil moisture, productivity and respiration. I'd argue ELM and other land models do fine with temperature, but really struggle with the soil moisture at seasonal and inter annual time scales. The changes to soil physics (the Hansen model) seem to improve dynamics of soil moisture stress, but without appropriate effects on the biogeochemistry. What modification could be made to the formulation of btran and xi_w? Are these functions that need new parameterizations, new forms, or that should be replaced in future generations of the model. I'm not asking for the simulations in the paper, but can these ideas be explored more in the discussion?

Response: The reviewer provided a great suggestion. We agree that after improving the simulation of SWP, the moisture scalars (btran and $\xi_W$) may also influence the biogeochemical processes. For example, no matter which SWP simulations were used, the ELMv0 had smaller interannual variability than the observations (Fig. 2). Specifically, the model was not able to capture the steep decreases in GPP and SR in the extreme drought year (i.e., 2012). These results indicate that the current model structure is not sensitive enough to environmental changes.

Thanks to the reviewer's comments, we have added more discussion of the possible influences of moisture scalars and possible improvements in the revision (Page 14, line 10 – 18):

> *"The calculation of the moisture scalars (e.g., btran and $\xi_W$) using empirical equations from SWP may be another potential reason for the insensitivity. For example, observational results have shown that there may be an optimal moisture point at which soil respiration peaks with significant reductions in*

*decomposition towards both dryer and wetter conditions (Linn and Doran, 1984; Franzluebbers, 1999; Monard et al., 2012; Sierra et al., 2017). In the ELMv0, however, the moisture scalar increases from 0 to 1 with the increase in soil moisture and does not decrease afterwards (Eq. 9). Thus, the ELMv0 may not be sensitive to extreme wet conditions. The linear empirical equation between the lower and upper thresholds ($\Psi_{min}$ and $\Psi_{max}$) may not capture non-linear moisture behaviours, leading to insensitive responses of biogeochemical processes to moisture change. Incorporating more mechanistic moisture scalars may improve the sensitivity of the model in response to moisture changes (Ghezzehei et al.; Yan et al., 2018)."*

Towards this end, it's still a little confusing what the most tractable way will be to parameterize soil hydrology in ELM. The authors discuss why the current approximation is sub-optimal (page 11-12), but can Hansen be applied in the global model? I'm assuming part of the motivation for this paper is to document the Hansen parameterization for work moving forward. If so, can the discussion more explicitly state how existing data can inform the global scale parameterization, application, and evaluation of the Hansen model? The current text seems somewhat vague.

Response: The reviewer asked an insightful question "*can the Hanson model be applied in the global model?*" and suggested to "*more explicitly state how existing data can inform the global scale parameterization, application and evaluation of the Hanson model*". We appreciate these suggestions.

The answer to the question is not a simple "yes" or "no". As the reviewer stated in an earlier comment, the ELM and other global models "*struggle with the soil moisture*" simulation. To improve the global simulations of SWP and SOC, two important aspects, an updated database and a well parameterized SWP-VWC relationship, are needed.

On the global scale, the ELMv0 used a SWP-VWC relationship which was parameterized by grouping over 1,000 soil samples into 11 USDA soil textural classes. The midpoints of sand and clay content in the 11 textural classes were used to extrapolate the relationship into the global scale. One potential issue is that soil samples in the same textural classes can have different sand and clay contents and SWP-VWC relationships, which may not be fully represented when they are grouped together. To address this issue, an updated SWP-VWC database with actual sand and clay content measurements (e.g., the UNsaturated SOil hydraulic Database, UNSODA), may enable improved relationships between model parameters and soil texture in the water retention models.

Meanwhile, different empirical models have been developed to describe the SWP-VWC relationship as shown in Table 1 and Fig. 1. These models could be evaluated with the UNSODA database, and the selected best-fit model(s) could be used to calculate SWP in the field from continuously monitored VWC (e.g., from the AmeriFlux network) on different spatial and temporal scales. Additionally, the database could also be used as a benchmark to evaluate simulations of soil water and biogeochemical processes in ESMs.

In the manuscript, we have discussed these points in Page 13 line 1 – 16:

*"Parameters in the default Clapp & Hornberger model used in the ELMv0 were derived from synthesizing data across soil textural classes (Clapp and Hornberger, 1978; Cosby et al., 1984; Lawrence and Slater, 2008). The data were derived from over 1,000 soil samples from 11 USDA soil*

*textural classes (Holtan et al., 1968; Rawls et al., 1976). The dependence of model parameters on soil texture were derived from a regression of these 11 data points, i.e., the mean parameter values of 11 soil textural classes against the sand or clay fractions (Cosby et al., 1984). Because no actual sand or clay content of soil samples was reported in the original databases (i.e., only the soil textural classes were reported), the sand and clay fractions used for the regression were obtained from midpoint values of each textural class (Clapp and Hornberger, 1978; Cosby et al., 1984). One potential issue is that soil samples in the same textural classes can have different sand and clay contents and SWP-VWC relationships, which may not be fully represented when they are grouped together. An updated SWP-VWC database with actual sand and clay content measurements could provide improved empirical relationships between model parameters and soil texture in the water retention model.*

*In addition, different empirical models have been developed to describe the SWP-VWC relationship (Brooks and Corey, 1964; Clapp and Hornberger, 1978; van Genuchten, 1980; Fredlund and Xing, 1994; Hanson et al., 2003). These models could be evaluated against data, and the selected best-fit model(s) could be used to calculate SWP in the field from continuously monitored VWC (e.g., from the AmeriFlux network) on different spatial and temporal scales. The database could also be used as a benchmark to evaluate simulations of soil water and biogeochemical processes in ESMs."*

It seems like 'fixing' a soil moisture bias in ELM exposed a compensating bias in the canopy photosynthesis (red lines in Fig. 4). I wonder if the proposed modifications to plant physiological parameters are also consistent with observations? Specifically, the add-hoc tuning (Table 2) of SLA (to reduce LAI) and FLRN (to modify Vcmax and maintain GPP with a less leafy canopy) certainly worked (Fig. 4). I wonder if the modifications are at all consistent with leaf physiology measurements from the site or from similar plants in the TRY database?

Response: The parameter adjustments followed a surrogate-based global optimization using measurements of C and energy fluxes at the site (Lu et al., 2018). The TRY database showed that the SLA for broadleaved deciduous forest ranges from < 0.005 to > 0.05 $m^2$ $g^{-1}$ C, with mean values of 0.015 $m^2$ $g^{-1}$ C (Kattge et al., 2011), which is similar to our modification (i.e., 0.01 $m^2$ $g^{-1}$ C). Similarly, the adjustment of the parameter *flnr* fell within the range of observations in the TRY database.

Minor and technical concerns:
In general, I'm not a fan of introducing new analyses / results at the end of the discussion (Fig. 5). As such, I'd encourage moving this display item into the results (section 3) and mentioning the experiment in the methods (section 2), but leave this up to the authors & editor to decide.

Response: We appreciate the suggestion for the manuscript presentation. We have added descriptions of Fig. 5 in Methods and moved Fig.5 to Results:

Page 6 line 21 – 22: *"The contributions and autotrophic and heterotrophic respiration to total SR were also calculated."*

Page 10 line 13 – 15: *"The contributions of autotrophic and heterotrophic respiration to total SR had a seasonal cycle (Fig. 5). The contribution of heterotrophic respiration to total SR ranged from 60% to 90%."*

Why not put the uncertainty bands on both parts of Fig. S5 (and other bar charts in the manuscript)?

Response: We have added the uncertainty bands in Fig. S5 and other bar charts in the revision.

References:
Carvalhais, N., Forkel, M., Khomik, M., Bellarby, J., Jung, M., Migliavacca, M., et al. (2014). Global covariation of carbon turnover times with climate in terrestrial ecosystems. Nature, 514(7521), 213-217. doi: 10.1038/nature13731.

Response: Thanks for the information.

Reviewer 2

Review: Evaluating the E3SM Land Model Version 0 (ELMv0) at a temperature forest site using flux and soil water measurements
Authors: Liang, Wang, Ricciuto, Gu, Hanson, Wood, Mayes
Synopsis: The authors run an out-of-the-box simulation of E3SM (which is the same as CLM4.5-CN; I'm not sure what makes E3SM distinct) and compare results to Eddy Covariance (EC) fluxes and soil moisture observations. The default model is found to have Gross Primary Productivity (GPP) and Soil Respiration (SR) that are too low when compared to observations. Near-surface Soil Water Potential (SWP), calculated using relationships that determine SWP from Volumetric Water Content (VWC) based on Clapp&Hornberger result in potentials that are too high during the winter, and too low in the summer. Overall, SWP is too low at low moisture in the near- surface soil, and was slightly high at depth when moisture content was higher. Five different treatments for relating VWC-SWP were tested, and the model of Hanson was found to have the smallest errors when compared to observations. However, when C&H was replaced with Hanson GPP was slightly low and SR slightly high when compared to observations, and the model did not reproduce either the amplitude or sign of interannual variability. Therefore, coefficients influencing Specific Leaf Area (SLA), fractional leaf N used in Rubisco and several coefficients controlling leaf senescence were changed, and results were improved in evaluation of mean seasonal cycles of LAI, GPP and SR.
Finally, there was speculation about which mechanisms and processes might be responsible for model-data mismatch after the aforementioned tuning was complete. These include model Q10 for heterotrophic respiration, microbial biomass and seasonality, and macroinvertebrate (earthworm) influence on carbon cycle processes, and root exudates. The authors exhort the community to pay particular attention to SWP in simulations, and to consider inclusion of these added processes in models.

Response: We appreciate the precise summary.

Review: One could consider this a model-tuning paper. A default model was run, deficiencies were noted, and changes were made to parameters and model physics. This is fine, and has been done many times previously (e.g. Sellers et al. 1989), but I'm not sure that the present paper really tells me anything about how the world works. I work with models that simulate land-atmosphere interaction, and there is nothing in this paper that makes me want to look at my model code and start performing tests and making changes. Hanson worked better, but it worked better at one place on the planet, at a particular deciduous forest (DBF) in the North American midwest. I suspect that if we were to perform evaluations like this at multiple EC sites (across

multiple DBF sites and across multiple PFTs), I expect that we would find that each of the VWC-SWP treatments would come out on top at least one or more times. We would also likely find that the SLA, Nitrogen and senescence parameters could take multiple values as well.

Response: We agree that the performance of the five SWP-VWC relationships may vary across different ecosystems. We chose the Hanson model because it performed the best at the study site. The purpose of this manuscript was to evaluate the ELMv0 using long-term site-level observations. As the reviewer mentioned, similar works have been widely done to improve the model performance.

We are sorry that the manuscript did not excite the reviewer. We are not sure which models the reviewer works with, but previous model evaluation papers show that the hydrological modification on biogeochemical cycles is a big issue in current land models (Todd-Brown et al., 2013; Carvalhais et al., 2014). In our manuscript, we have attempted to explore this issue with the ELMv0 and the long-term observations at the MOFLUX site. One of the most important points is that the ELMv0 is not able to simulate SWP well, and a better representation of SWP, using site-specific data if available, can significantly improve the mean annual simulations of biogeochemical processes. Because many current efforts involving soil carbon cycle modeling are focused on better modeling the effects of changes in temperature, and whether or not to include explicit microbes, we felt it was important to point out the role of soil moisture on model performance. We also agree with the reviewer that the SLA, nitrogen and senescence parameters may change across sites. Common parameter values based on biome and/or soil types would be a better choice if modeling larger spatial scales.

My main complaint now levied, I will also say that just because this paper does not particularly excite me, there is nothing wrong with the analysis. The paper follows a logical progression, and the analysis and presentation of results is done professionally and is easy to follow. I think there is value in the paper, and my official recommendation is to accept this manuscript with minor revisions.

Response: We appreciate the positive comments on the manuscript.

The paper is quite short (14 pages), which is nice, but I think there might be some expansion of analysis and explanation that would add value to the research.

Merely stating that the model was unable to capture observed response to the 2012 drought is extremely unsatisfying. This is an opportunity to explore model behavior, and perhaps gain valuable insight into processes and mechanisms. I find it interesting that observed near-surface (10cm) SWP (Fig 3b) was not exceptionally low in 2012; the year did not look much different from 2011 or 2013, and in fact looked wetter than 2005-2007. That is interesting; what was deep soil SWP doing in those years? From Figure 2, we see that observed SR oscillated up and down between 2005-2007, while GPP dropped from 2005 to 2007. What was the model doing? What did BTRAN look like in 2012, as compared to other years? How about LAI? Is there a near-surface water table in the simulations that prevents root stress? Are there constraints on stomatal conductance due to high VPD or unfavorable temperature? What did they do in 2012? In fact, simulated SR and GPP both increased from 2011 to 2012, while there were dramatic drops in the

observations of both. If the model does not respond to the drought, you should be able to tell your reader why, and speculate whether that behavior is realistic or not, and how that behavior might impact model performance in other years. I would like to see some exploration of IAV, and explanation of why Hanson provides an upgrade from C&H in this regard.

Response: We appreciate the valuable comments, especially the suggestion of explorations of 2012 drought and interannual variability. In the manuscript, we have an entire section (i.e., section 4.2) to discuss the representation of seasonal and interannual variabilities in the ELMv0. We explored the interannual variability from several aspects.

First, to answer the reviewer's questions, we have conducted more analyses. It was an extremely dry year in 2012, with much lower precipitation than other years (Fig. S9 and also see below). However, neither the upper nor the deeper layer SWP in 2012 was extremely low as well (Fig. S10 and also see below). This could be because that the severe drought-pathogen interactions in 2012 resulted in a significant stem mortality of tree species (Wood et al., 2017). Thus, the observed steep decreases in GPP and SR are likely because of the species mortality. In addition, the stem mortality led to lower moisture loss through evapotranspiration (Fig. S9), resulting in no soil moisture decrease. In the model, however, the water limitation to biogeochemical cycles is primarily controlled by soil water stress. In other words, the ELMv0 had moisture modifications at the physiological level, but not at the plant community level. As a result, although the better presentation of SWP improved the mean annual simulations of biogeochemical processes, the model was not able to capture the mortality and the interannual variability of GPP and SR.

[Figure]

**Figure S9. Mean annual precipitation and evapotranspiration at the MOFLUX site from 2005 to 2013.** Both precipitation and evapotranspiration were lower in 2012 than other years. The grey bars show the multi-year standard deviation.

[Figure]

**Figure S10. Mean annual soil water potential (SWP) at 10 cm and 100cm at the MOFLUX site.** The grey bars show the multi-year standard deviation.

Second, the calculation of the moisture scalars (e.g., btran and ξw) using empirical equations from SWP may be another potential reason for the insensitivity. For example, observational results have shown that there may be an optimal moisture point at which soil respiration peaks with significant reductions in decomposition towards both dryer and wetter conditions (Linn and Doran, 1984; Franzluebbers, 1999; Monard et al., 2012; Sierra et al., 2017). In the ELMv0, however, the moisture scalar increases from 0 to 1 with the increase in soil moisture and does not decrease afterwards (Eq. 9). Thus, the ELMv0 may not be sensitive to extreme wet conditions. The linear empirical equation between the lower and upper SWP thresholds ($\Psi_{min}$ and $\Psi_{max}$) may not capture non-linear moisture behaviours, leading to insensitive responses of biogeochemical processes to moisture change. Incorporating more mechanistic moisture scalars may improve the sensitivity of the model in response to moisture changes (Ghezzehei et al.; Yan et al., 2018).

Additionally, lacking representations of microbial and macroinvertebrate dynamics may also potential reasons for the low seasonal and interannual variability in the model. We also discussed that the temperature sensitivity was unlikely a reason.

Please see Section 4.2 for more details. We have made revision in the Abstract and Conclusions.

Related to the above is the fact that in 2007 there was a significant drop in observed SR and GPP when compared to 2006. The default model (C&H) showed drops that are more similar to the amplitude of the observed reduction, even if there is an offset or bias. In fact, the Hanson model shows almost no interannual variability (IAV) in SR and GPP at all. Is this really an improvement? One might make the case that you would have a better simulation of the observed flux by increasing Vcmax (and perhaps SLA and the senescence parameters) in CLM without changing the soil; your GPP would go up, which would translate into larger carbon pools and subsequent increased RS. You would also retain a more realistic comparison with observed IAV. Is there a reason to suspect that this would not work?

Response: The study site experienced drought in 2007, which also resulted in an increased tree mortality though it was milder than that in 2012 (Wood et al., 2017). As discussed above, the model was not able to capture the mortality and the interannual variability without vegetation and microbial dynamics. Therefore, although the default model showed drops that were more similar to the amplitude of the observed reduction in 2007, it was because of the wrong reason (i.e., wrong SWP).

The reviewer asked "*is this really an improvement*". The default model did not simulate SWP, GPP and SR well. We improved the modeled SWP by better representing the SWP-VWC relationship in the model. The better representation of SWP further improved the mean annual simulations of GPP and SR. We think this is an improvement because the changes in GPP and SR were because of the right SWP simulation.

The reviewer also asked if we suspect changing Vcmax would not work. We changed the SWP because we found the model was not able to simulate it properly when comparing with observations. For Vcmax, we do not have any observational evidence that it is wrong in the model.

Although the model did not match the observed interannual variability, the appropriate simulation of SWP led to improved mean annual simulations of GPP and SR. Based on that, we think better representations of vegetation dynamics, moisture function, and microbial activity may address the issue of interannual variability. We have added more discussion in the revision, thanks to the reviewer's comments.

In section 4.1 two paragraphs (lines 30-33 on page 11, continued in lines 1-7 on page 12; lines 8-13 on page 12) describe how C&H was developed from textural classes and not sand/clay fractions, and how models might make use of near-continuous SWP observations. I'm not sure what these 2 paragraphs bring to the analysis, since neither has been done. Do they merit this much attention?

Response: We think these two paragraphs are necessary. The default Clapp & Hornberger model failed to simulate the SWP in the ELM. An obvious question to ask was why it failed and how to improve. We have used these two paragraphs to discuss this issue. The default Clapp & Hornberger model's parameters were dependent on soil sand/clay fractions. However, it used data without actual measurements of soil sand/clay fractions, leading to inaccurate simulations. As a result, we propose that an updated SWP-VWC database with actual sand and clay content measurements may enable improved relationships between model parameters and soil texture in the water retention model.

Different empirical models have been developed to describe the SWP-VWC relationship (Brooks and Corey, 1964; Clapp and Hornberger, 1978; van Genuchten, 1980; Fredlund and Xing, 1994; Hanson et al., 2003). As the reviewer mentioned above, the performance of different SWP-VWC relationships may vary across different ecosystems, which we totally agree. Thus, it may ultimately be necessary to evaluate these models using a global database and select the best-fit model(s) on the global scale.

In summary, we think these two paragraphs are needed from the perspective of model evaluation and improvement.

The authors state that "SWP in simulations in ESMs should be calibrated carefully with observations...", but this is clearly impossible and unrealistic in global simulations. If the ultimate goal is correct simulations of global biogeophysical behavior, then we have a disconnect between what the authors are doing here (tuning at a single site) and what we are told is the ultimate goal (accurate representation of global carbon cycle). This is a persistent and real problem. We calibrate our models on site-level data and then extend that behavior to the globe. I'd be interested in some discussion of how we might use site-level studies to improve global simulations.

Response: We appreciate the suggestion of discussing how to improve global simulations using site-level studies. It is true that there is no global grid-based SWP database. However, the research community has collected paired measurements of VWC and SWP, as well as soil characteristics, in a variety of soil types and ecosystems (e.g., the UNSODA database). These data can be used to calibrate SWP-VWC relationships and SWP simulations in models. There are many sites, such as the MOFLUX site in this study, collecting long-term hydrological and biogeochemical data. These data are very useful to evaluate whether improving the SWP simulation is one of the right reasons for model improvement.

Thanks to the reviewer's suggestion, we have added more discussion in the revision (Page 12 line 20 – 27):

> *"Our analyses in this study indicate that improving the modelled SWP can significantly improve mean annual GPP and SR simulations. Thus, we propose that the SWP simulation in ESMs should be calibrated carefully with observations, and/or by using different model representations of the SWP-VWC relationship. Because there is no global grid-based SWP database, paired measurements of VWC and SWP are needed along with soil characteristics in a variety of soil types and ecosystems. These data can be used to calibrate SWP-VWC relationships and SWP simulations in models. Besides, there are many sites, such as the MOFLUX site in this study, collecting long-term hydrological and biogeochemical data. These data are useful to evaluate whether better SWP simulation will improve biogeochemical cycling simulations."*

Specific Comments:
• Equation 4: The subscript should be liveCroot, should it not?

Response: Yes. We have corrected it.

• Page 4, line 9: coarse

Response: Revised (Page 4, line 9).

• Page 5: is 'residual' water content the same as wilt point?

Response: The residual water content is not exactly the same as the wilting point. The wilting point, by definition, is the minimal point of soil moisture the plant requires not to wilt. It is the water content at -1.5MPa of suction pressure. On the other hand, the residual water content is a parameter to determine the shape of the SWP-VWC relationships. As summarized by Vanapalli et al. (1998), the definition of residual water content varies depending on the SWP-VWC model, but it can exceed the wilting point (i.e., at -1.5MPa of suction pressure).

• Table 1: I'm not sure what AIC is: shouldn't it be explained, even if briefly?

Response: The Akaike Information Criterion (AIC) is an estimator of the relative quality of models for a given set of data. The smaller AIC value, the better. It was calculated by

$$AIC = a ln\left(\frac{\sum(\hat{\varepsilon})^2}{a}\right) + 2b$$

We have added the AIC equation in the Method section (Page 6, line 7 – 11).

• Page 8, line3: I know what 'btran' is, but some of your readers may not. You should explain this variable.

Response: We have added explanation in the revision (Page 8, line 12 – 14):

*"The btran is the transpiration beta factor, which controls the soil water limitation to transpiration and photosynthesis, while $\xi_w$ is the soil moisture modifier for heterotrophic respiration as shown in Eq. (9)."*

• Figure 4, Figures S6-S7: Make lines darker, shading lighter. Hard to discern individual simulations.

Response: Revised as suggested.

• I might have missed this, but what is the porosity and sand/clay content of the soil at the MOFLUX site? If VWC at depth regularly drops to between 15-20% (Fig 1b) then it must have considerable sand content. My recollection of more clayey soils is that wilt point will be much higher. Is this soil representative of the region and/or PFT?

Response: The dominant soils are the Weller silt loam and the Clinkenbeard very flaggy clay loam (Young et al., 2001). This soil is representative of the region. The sand and clay contents in the lower layer (Fig. 1b) are 8.66% and 39.89%, respectively. According to its definition (i.e., at -1.5MPa of suction pressure), the VWC at the wilting point of the lower layer is approximate 23% (Fig. 1b). The values shown in Fig. 1 were not measured in the field. Instead, to derive the soil water retention curves, soil samples were collected in the area of the flux tower base at two depths: 0 to 30 cm and below 30 cm. Samples were evaluated periodically for soil water potential using a dewpoint potentiometer (Decagon Devices, Model WP4C) as they dried over time (Hanson et al., 2003). Thus, the measurements went beyond the wilting point.

• Equation 9: It appears that the environmental modifier for water has value of 0 and low water (conditions too dry for microbial activity), varies between 0-1 for moisture up to (PSI)max. What is the difference between (PSI)max and (PSI)s? Are they the same? Most models I am familiar with will have an 'optimum' soil water content or potential for respiration, the idea being that either too dry or too wet (anaerobic) conditions are unfavorable for microbial decomposition of carbon stocks. The 'too wet' does not seem to be the case here. Why is that?

Response: We apologize for the typo. $\Psi_s$ should be $\Psi_{max}$, which is the matric water potential under saturated conditions. The issue has been fixed in the revision (Page 4, Eq. 9 and line 20).

The reviewer asked a very interesting question why the moisture modifier ($\xi_w$) did not decrease under "too wet" conditions. In the ELM, the moisture modifier was based on observational data (Orchard and Cook, 1983; Andren and Paustian, 1987). These data were produced a few decades ago. However, as the reviewer mentioned, more recent data showed there might be an optimal moisture point at which soil respiration peaks with significant reductions in decomposition towards both dryer and wetter conditions (Linn and Doran, 1984; Franzluebbers, 1999; Monard et al., 2012; Sierra et al., 2017). As a result, the calculation of the moisture scalars may be a

potential reason for the unrealistic simulated interannual variability, which we discussed in the revision (Page 14, line 10 – 18):

> *"The calculation of the moisture scalars (e.g., btran and $x_W$) using empirical equations from SWP may be another potential reason for the insensitivity. For example, observational results have shown that there may be an optimal moisture point at which soil respiration peaks with significant reductions in decomposition towards both dryer and wetter conditions (Linn and Doran, 1984; Franzluebbers, 1999; Monard et al., 2012; Sierra et al., 2017). In the ELMv0, however, the moisture scalar increases from 0 to 1 with the increase in soil moisture and does not decrease afterwards (Eq. 9). Thus, the ELMv0 may not be sensitive to extreme wet conditions. The linear empirical equation between the lower and upper thresholds ($\Psi_{min}$ and $\Psi_{max}$) may not capture non-linear moisture behaviours, leading to insensitive responses of biogeochemical processes to moisture change. Incorporating more mechanistic moisture scalars may improve the sensitivity of the model in response to moisture changes (Ghezzehei et al.; Yan et al., 2018)."*

• Using 10 years of tower forcing to perform a 200-year spinup of carbon pools concerns me. I understand that this might be all the tower data available, but, especially for carbon pools, I'm concerned that anomalies in the 10-year meteorology may be aliased onto pool size. Did you consider using a reanalysis product (CRU, NCEP, ECMWF) for spinup and then use the tower data for the transient run?

Response: We used preindustrial atmospheric forcing (e.g., $CO_2$ concentrations and nitrogen deposition) and site-specific meteorological measurements to perform the spinup. We hope the reviewer would agree that both tower measurements and reanalyzed products will produce biases and uncertainties when they are used for the spinup. The resolution mismatch and algorithm uncertainty of reanalyzed products may significantly influence the site-level results. In addition, using the reanalyzed data first followed by tower data may produce abrupt changes in the model simulation. For the site-level study, the *in situ* measurements were our first choice. Using site-specific meteorological measurements to perform spinup has been applied in many studies (e.g., Mao et al., 2016; Duarte et al., 2017). Thus, we chose to use the site-specific meteorological measurements to perform both the spinup and transient simulations.

[revised manuscript text omitted]

---

## Author Response (AR3)

**Letter of Response**

Dear Dr. Tomomichi Kato

We are re-submitting the manuscript entitled "Evaluating the E3SM Land Model version 0 (ELMv0) at a temperate forest site using flux and soil water measurements" to be considered for publication in *Geoscientific Model Development*. We appreciate the opportunity to re-submit the manuscript, which has been thoroughly revised based on the reviewer's comments. We are submitting a letter of point-by-point responses to reviewers' comments, as well as a marked-up revision. We hope you will find our revision thorough and satisfactory.

Sincerely,
Junyi Liang
* * *
Author Note: The reviewers' comments are in black and the responses follow in blue. Page numbers and line numbers in the responses are those in the marked-up revision following the response letter.

Review: Liang et al., MOFLUX site, soil respiration

In general, the authors have addressed my main points from the initial review. This paper can be published, with the minor comments from below addressed. I do not need to see this paper again.

Response: We appreciate all the reviewer's comments, which are very helpful to improve the manuscript.

Specific comments:
Page 6, last line: It looks to me like C+H overestimates SMP when moisture is ***below*** 15% for depth below 30cm.

Response: All measured volumetric soil water content (VWC) for the soil water retention curve for the depth below 30 cm was greater than 15% (x-axis in Fig. 1b). Within the measured VWC range (i.e., 15% to 40%), the default ELM (which is *C+H* in the reviewer's comment) overestimated soil water potential (Fig. 1b). Thus, in the previous version, we described the results as "*For soil below 30 cm, the ELMv0 showed a consistent overestimation of SWP when VWC exceeded 15%*".

The reviewer's comment suggests that the sentence may result in misunderstanding. In the revision, we have rephrased the sentence as (page 6, line 28):

   *"For soil below 30 cm, the ELMv0 showed a consistent overestimation of SWP (Fig. 1b)."*

Page 10, line 4: There is no range on the GPP plot of Figure S5. Also, it looks like SR and GPP plots are reversed (a and b) when compared to their referencing in the text.

Response: We have added the range on the GPP panel and shifted the two panels corresponding to the citation in the text. Please see the revised Fig. S5.

Page 12, line 5: "First, the Hanson Model significantly increased GPP." This is an incorrect statement. The Hanson model does not calculate GPP. The Hanson model determines SMP, and SMP is related to GPP through the ability of the model to capture soil moisture for use in transpiration. Interestingly, the Hanson SMP is only higher than the C+H SMP in the top 30cm of soil when VWC is < 15%. Everywhere else (top 30 cm, VWC > 15%, all VWC below 30cm) the Hanson SMP value is below (larger negative) than the C+H (Figure 1). What this says to me is that ELM transpiration (and GPP) is critically dependent on soil moisture in the upper 30 cm of the soil, and this upper soil is frequently at VWC below 15%. There should be some discussion of this. Do you think this result is realistic? What this is saying is that trees in the American Midwest are not at all dependent upon soil moisture below 30cm in the soil. Do you believe that to be true? Myself, I find this result suspicious. I was under the impression that deeper roots are critical to tree survival. This result contradicts that, as Hanson SMP was lower than C+H at all VWC below 30cm.

Response: We have revised the sentence as (page 11, line 14-15):

*"First, the changes in SWP with the Hanson model increased plant transpiration and GPP in the model."*

The reviewer suggested to discuss why the ELM transpiration is critically dependent on soil moisture in the upper 30 cm. The reviewer also observed that "*the Hanson SMP is only higher than the C+H SMP in the top 30cm of soil when VWC is < 15%. Everywhere else (top 30 cm, VWC > 15%, all VWC below 30cm) the Hanson SMP value is below (larger negative) than the C+H*". We greatly appreciate the valuable comments. In temperate forests, approximate 60% of plant roots are distributed in the upper 30 cm of the soil (Jackson et al., 1996). That means the moisture in the upper 30 cm is critically important for transpiration and GPP. Therefore, we agree with the reviewer's observation that "*ELM transpiration (and GPP) is critically dependent on soil moisture in the upper 30 cm of the soil*". This suggests that the ELM was able to represent the dependence of transpiration (and GPP) on soil moisture change. In addition, one important trend at the MOFLUX site is that soil moisture is lower during the peak growing season than other times (Fig. 2). As a result, the improved SMP simulation by the Hanson model during the peak growing season when VWC < 15% in the upper 30 cm of the soil played a critical role for the improvement of GPP and SR simulations.

To respond to the reviewer's comments, we have added discussion in the revision (page 12, line 2-6):

*"In addition, the improvement of GPP and SR simulations was primarily due to the better simulation of the SWP in the upper 30 cm of the soil, as approximately 60% of plant roots are distributed in the upper 30 cm of the soil in temperate forests (Jackson et al., 1996). One important trend at the MOFLUX site was that soil moisture was lower during the peak growing season than during other times. As a result, the improved SWP simulation in the upper 30-cm soil during the peak growing season played a critical role in the improved simulation of GPP and SR."*

Section 4.2: The lack of IAV in the model may be tied to the dependence on near-surface soil moisture. It appears that the main change imposed by using the Hanson model is that SMP is higher, so moisture is more readily available to roots, in the upper 30cm of the soil, and only when VWC is below 15%. I would expect that VWC in the upper 30cm gets reduced to very small amounts every year. In that case the model would not be expected to see much IAV, ever.

So it may be that the lack of IAV in the model has nothing to do with mortality rates or pathogens, and everything to do with how the model extracts water from the soil. On the other hand, Figure S10 shows that SWP was lowest during the entire record in the period 2005-2007, and the GPP in those years was not as extremely depleted as it was in 2012 (Figure 2). This is something the authors should discuss.

Response: We appreciate the valuable comments. We agree that the dependence of GPP and SR on upper layer soil moisture may be one of the reasons for the lack of inter-annual variability. In addition, the long-term measurements suggest that failing to capture the change at the community level, such as mortality and drought-pathogen interactions, may also contribute to the inter-annual variability.

The reviewer also suggested to discuss why "*SWP was lowest during the period of 2005-2007, but the GPP in those years was not as extremely depleted as it was in 2012*". Field inventory data at the study site showed that the severe drought-pathogen interactions in 2012 resulted in a significant stem mortality of tree species (Wood et al., 2017). Thus, the observed steep decreases in GPP and SR could be because of the species mortality. The stem mortality led to lower moisture loss through evapotranspiration (Fig. S9), resulting in no soil moisture decrease (Fig. S10).

In the manuscript, we have discussed possible reasons regarding the lack of inter-annual variability in the model simulation (page 13, line 20-33).

[revised manuscript text omitted]